# A juxtamembrane basolateral targeting motif regulates signaling through a TGF-β pathway receptor in *Drosophila*

Aidan J. Peterson[1], Stephen J. Murphy[2], Melinda G. Mundt[1], MaryJane Shimell[1], Edward B. Leof[3], Michael B. O'Connor[1] *

1 Department of Genetics, Cell Biology & Development and the Developmental Biology Center, University of Minnesota, Minneapolis, Minnesota, United States of America, 2 Center for Individualized Medicine, Mayo Clinic College of Medicine, Rochester, Minnesota, United States of America, 3 Thoracic Diseases Research Unit, Department of Pulmonary and Critical Care Medicine, Mayo Clinic College of Medicine, Rochester, Minnesota, United States of America

* moconnor@umn.edu

**Data Availability Statement:** All relevant data are within the paper and its Supporting Information files.

## Abstract

In polarized epithelial cells, receptor–ligand interactions can be restricted by different spatial distributions of the 2 interacting components, giving rise to an underappreciated layer of regulatory complexity. We explored whether such regulation occurs in the *Drosophila* wing disc, an epithelial tissue featuring the TGF-β family member Decapentaplegic (Dpp) as a morphogen controlling growth and patterning. Dpp protein has been observed in an extracellular gradient within the columnar cell layer of the disc, but also uniformly in the disc lumen, leading to the question of how graded signaling is achieved in the face of 2 distinctly localized ligand pools. We find the Dpp Type II receptor Punt, but not the Type I receptor Tkv, is enriched at the basolateral membrane and depleted at the junctions and apical surface. Wit, a second Type II receptor, shows a markedly different behavior, with the protein detected on all membrane regions but enriched at the apical side. Mutational studies identified a short juxtamembrane sequence required for basolateral restriction of Punt in both wing discs and mammalian Madin-Darby canine kidney (MDCK) cells. This basolateral targeting (BLT) determinant can dominantly confer basolateral localization on an otherwise apical receptor. Rescue of *punt* mutants with transgenes altered in the targeting motif showed that flies expressing apicalized Punt due to the lack of a functional BLT displayed developmental defects, female sterility, and significant lethality. We also show that apicalized Punt does not produce an ectopic signal, indicating that the apical pool of Dpp is not a significant signaling source even when presented with Punt. Instead, we find that basolateral presentation of Punt is required for optimal signaling. Finally, we present evidence that the BLT acts through polarized sorting machinery that differs between types of epithelia. This suggests a code whereby each epithelial cell type may differentially traffic common receptors to enable distinctive responses to spatially localized pools of extracellular ligands.

**Funding:** MBO was funded by grant R35GM118029 for the National Institute of General Medicine. The funders had no role in study design, data collection and analysis, decision to publish, or preparation of the manuscript.

**Competing interests:** The authors have declared that no competing interests exist.

**Abbreviations:** AP, adaptor protein; A/P, Anterior/Posterior; BLT, basolateral targeting; Dpp, Decapentaplegic; IF, immunofluorescence; MDCK, Madin-Darby canine kidney; SJ, septate junction; TβRII, TGF-β Type II receptor; WT, wild-type.

## Introduction

Polarization of cells underlies core behaviors of cells and tissues by imparting directionality to important processes including adhesion, uptake, and secretion. A common structural motif in metazoans is the epithelium, a contiguous sheet of cells connected by cell–cell junctions. Epithelial cells are inherently polarized, typically possessing a basement membrane at the basal surface and an apical surface exposed to a lumen. The junctions provide compartmentalization by partitioning the basolateral and apical membrane domains within a cell and collectively form a barrier separating the basolateral and apical fluid pools surrounding the cell [1]. Beyond these common features, epithelia have diverse morphologies and molecular characteristics befitting their tissue type.

In the context of cell–cell signal transduction, polarized epithelia have access to regulatory schemes beyond the simple ligand-receptor paradigm. Because the ligand pools and the membrane domains are each divided into 2 independent partitions, there are conceptually 4 combinations of ligand and receptor availability in which effective signaling only occurs when exposed receptors can bind the ligand. Some signaling cascades originate at the apical surface, such as the Notch pathway wherein apical receptors bind to ligands in the lumen [2]. In other cases, ligand and receptor isolation conditionally limits signaling. For example, the heregulin-α ligand and its receptor are physically separated in intact airway epithelium but are exposed to each other upon wounding and thence signal to promote tissue repair [3]. A prominent signal transduction module associated with basolateral signaling is the TGF-β pathway. The canonical pathway consists of extracellular ligands that bind to transmembrane receptors, which transduce signals to intracellular Smad proteins to influence diverse processes [4]. Cell culture studies have found TGF-β receptors restricted to the basolateral membrane domains of several types of epithelial cells. This has been observed for the core Type I and Type II TGF-β receptors [5,6] and the Type III TGF-β receptor [7]. In Madin-Darby canine kidney (MDCK) epithelial cell culture, the functional consequence of TβRI and TβRII being restricted to basolateral membrane domains is that addition of exogenous ligand produces robust signaling only when applied to the basolateral fluid [8].

In a case featuring 2 pools of ligand, the localization of receptors could control the response to multiple, potentially overlapping signals in a tissue. Gibson and colleagues [9] described such a configuration in the *Drosophila* wing disc, where the Decapentaplegic (Dpp) ligand is detectable in 2 spatial pools: a graded stripe centered around the Anterior/Posterior (A/P) border of the wing pouch [10] and in the lumenal space between the wing pouch and the squamous peripodial membrane. This raised the question of how the graded domain can act as a morphogen if all cells are exposed to the lumenal Dpp pool. Given the reported basolateral restriction of mammalian TGF-β receptors, we hypothesized that specific receptor localization in the wing disc plays a role in signal transduction and tissue patterning.

We thus undertook a study of TGF-β pathway receptors in the *Drosophila* model system to determine if restricted basolateral localization occurs and how it impacts signaling. Three questions were experimentally addressed: Do receptors exhibit polarized localization? What is the *cis*-acting determinant for basolateral restriction? How does localization impact signal transduction? We find that membrane localization is qualitatively different for Type II receptors, with only Punt exhibiting tight basolateral localization in the wing disc. A *cis*-acting basolateral determinant from Punt that functions in insect and mammalian epithelial cells was mapped to the cytoplasmic juxtamembrane region. We show that membrane targeting controls signal output in the wing disc and that this determinant influences patterning and viability.

## Results

### *Drosophila* TGF-β/BMP receptors exhibit diverse membrane localization in wing disc epithelium

TGF-β family signaling is crucial for the growth and patterning of the wing disc. Most prominently, an extracellular Dpp protein gradient signals through Mad to pattern the tissue during larval development and promote proliferation [11]. The main receptors required for this signaling are Thickveins (Tkv, Type I) and Punt (Type II). The Gbb ligand and Saxophone receptor also contribute to BMP signaling in the disc [12]. The Activin branch has been reported to promote proliferation [13,14] and influence a subset of Dpp target genes [15,16]. As a first step to build a spatial model of signaling that incorporates epithelial polarity, we assessed the membrane location of key Dpp receptors in wing disc epithelial cells. The disc is composed of epithelial cells that form a sac with a lumenal space. We detected receptors in fixed tissues and compared membrane distribution to endogenous protein markers for junctions and the apical membrane domain (Fig 1A, S1 Fig). Tkv was found at both basolateral and apical positions, with some enrichment near the junctions (Fig 1B, S1 Fig). The same result was obtained with overexpressed Tkv and an endogenous reporter reagent (S1 Fig). Because the main Type I receptor does not exhibit restricted membrane positioning, we turned to the Type II receptors.

The Type II receptors Punt (encoded by *put*) and Wishful Thinking (Wit, encoded by *wit*) are both present in the wing imaginal disc [17,18], where Punt function is critical for wing development [19]. Overexpressed Punt protein exhibited basolateral restriction, with the bulk of the protein staying basal to the septate junction (SJ), which is marked by a concentrated stripe of FasIII or Dlg staining (Fig 1C). By contrast, Wit was enriched in the apical region overlapping aPKC, with clear but weaker staining along the basolateral membrane (Fig 1D). Orthogonal projections of confocal images through the wing disc confirm the differential localization of Punt-GFP and Wit-GFP (Fig 1E–1G). Parallel results were obtained with tagged constructs expressed at endogenous levels (Fig 1H and 1I). Because of the clear difference in membrane localization between these 2 functionally and structurally similar Type II receptors, we conducted further studies to explore the *cis*-acting determinants of their membrane targeting.

### The cytoplasmic domain of Punt directs basolateral restriction in evolutionarily distant epithelia

To coarsely map the targeting activity, we assessed the localization of chimeric proteins containing ectoplasmic and cytoplasmic regions of Type II receptors (Ectoplasmic:Cytoplasmic portions from the indicated protein) (Fig 2A). Upon expression in the wing discs of transgenic larvae, the Punt:Wit chimeric protein had similar distribution to the control Wit:Wit protein, with a clear apical enrichment over the aPKC stripe (Fig 2B and 2C). However, the Wit:Punt protein was basolaterally restricted, indicating that the cytoplasmic domain of Punt is responsible for localization (Fig 2D).

The mammalian TGF-β Type II receptor (TβRII) utilizes a basolateral targeting (BLT) motif in the cytoplasmic kinase domain [20,21]. Given the parallel membrane localization of fly and mammalian receptors, we expressed Punt in MDCK epithelial cells to test the conservation of protein targeting. Remarkably, Punt was targeted to the basolateral membrane domain (Fig 2F), revealing the action of a cell biology pathway operating on TGF-β family receptors in animals separated by over 500 million years of evolution [22]. To dissect the protein regions driving this localization, a series of progressive truncations was analyzed to detect a presumptive targeting motif (Fig 2E, S2 Fig). Surprisingly, removal of the sequence region

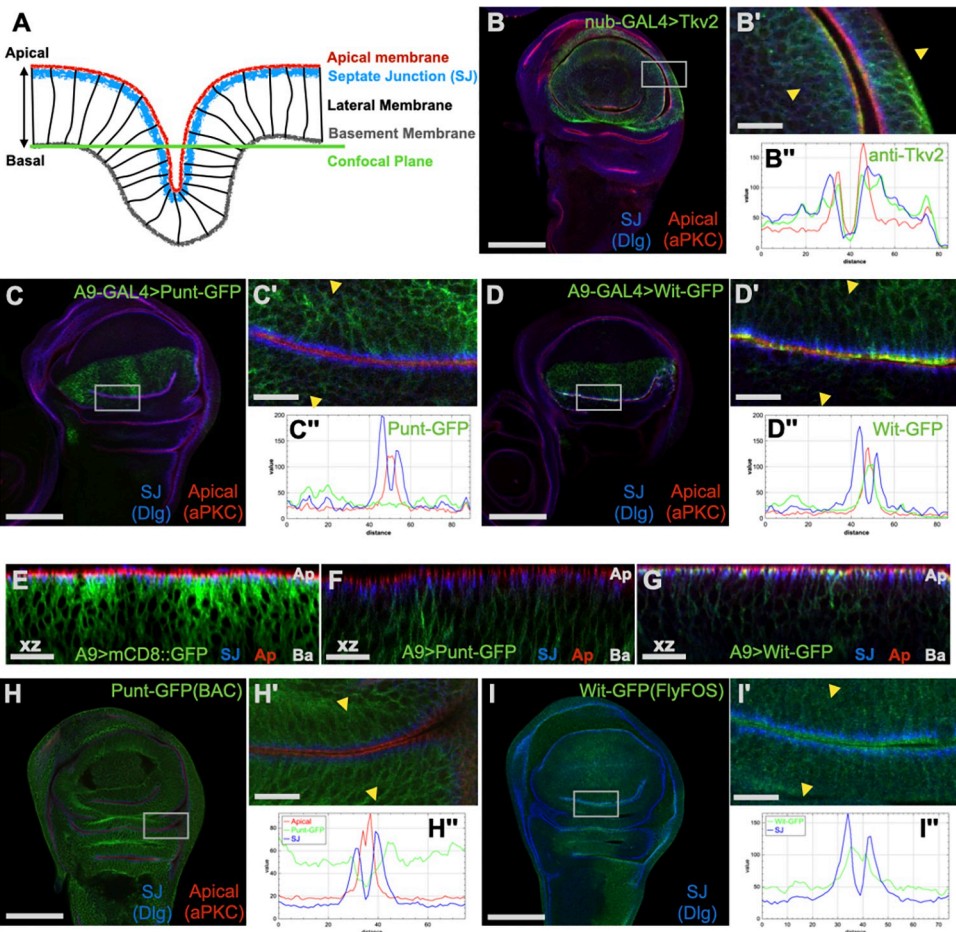

**Fig 1. Punt exhibits basolateral membrane localization in the wing disc. (A)** Schematic showing apical–basal features of the wing disc epithelium. Stereotypical folds bring 2 surfaces near each other with apical sides in apposition. Confocal images through the folds thus reveal the apical–basal polarity axis of the columnar epithelial cells. **(B)** Tkv expressed with *nub*-GAL4 was detected at the SJ (marked with FasIII or Dlg), as well as basal, lateral, and apical membranes (marked by aPKC stripe). Tkv2 was visualized by IF using an antibody specific to this isoform (B). See S1 Fig for additional Tkv localization tests. **(C–G)** GFP fluorescence signal conveys the membrane distribution of Type II receptors relative to junctions. Punt-GFP was largely excluded from the SJ (Dlg) and apical (aPKC) regions but detected at basolateral surfaces (C). Wit-GFP is enriched at the apical membrane but also detected basolaterally (D). **(E–F)** Apicobasal distribution of GFP-tagged proteins shown by xz projections of confocal z-sections. Apical (Ap) and Basal (Ba) sides are labeled. mCD8::GFP driven by A9-Gal4 was distributed through the columnar epithelial cells, with clear overlap at the SJ (E). Punt-GFP was excluded from the SJ and apical areas, displaying basolateral restriction (F), whereas Wit-GFP exhibited broad membrane localization with prominent overlap with the SJ and apical regions (G). **(H, I)** Endogenous-level tagged constructs confirm differential localization of Punt and Wit. Punt-GFP (H) or Wit-GFP (I) were detected by anti-GFP IF and analyzed as for Panels C and D; apical localization of Wit-GFP is inferred by staining apical to the SJ. Single prime (′) images are magnified views where 2 portions of the epithelium lean toward each other. Double prime (″) images are intensity profiles perpendicular to the line demarcated by the junction staining (yellow arrowheads in single prime panels; see S1 Fig for schematic of profile markers). Scale bars: 100 μm for B, C, D, H, I; 15 μm for B′, C′, D′, E, F, G, H′, I′. IF, immunofluorescence; SJ, septate junction.

corresponding to the mammalian "LTA" targeting motif [20] did not disrupt BLT (Fig 2G). Indeed, removal of nearly the entire cytoplasmic portion of Punt produced a protein with basolateral localization (Fig 2I). This behavior was recapitulated in a truncation series in the context of chimeric receptors containing variable lengths of the Punt cytoplasmic domain (S2 Fig). These data reveal that BLT of TGF-β pathway receptors in epithelia is phenomenologically conserved from flies to mammals, but that different portions of the protein and amino acid sequences direct the sorting of fly Punt and mammalian TβRII.

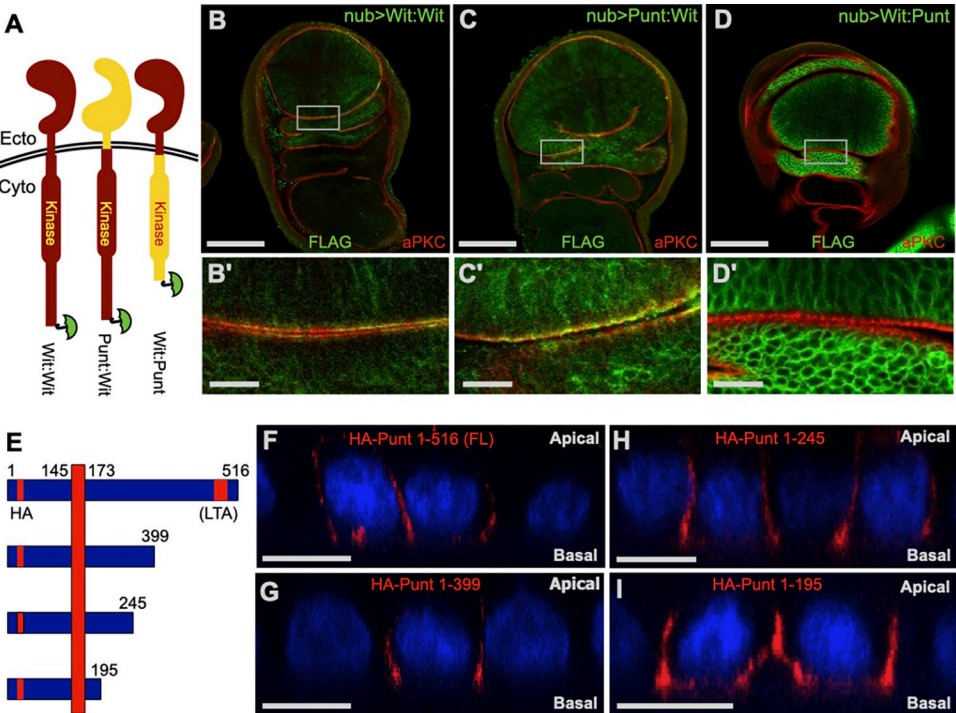

**Fig 2. Basolateral localization of Punt is conserved in MDCK cells and maps to a short region of the cytoplasmic domain. (A–D)** Chimeric proteins of Punt and Wit map the BLT activity to the cytoplasmic domain of Punt. The chimeric proteins comprise the extracellular and TM region (Ecto) and the cytoplasmic (Cyto) regions of the indicated proteins, with a carboxyl-terminal FLAG epitope (green umbrella symbol) for detection by IF. Wit:Wit staining overlapped with the apical marker (B), as did staining for the Punt:Wit chimera (C). TheWit:Punt chimera displayed basolateral distribution (D). Single prime (′) images show boxed areas at higher magnification. **(E–I)** Fly Punt expressed in polarized mammalian MDCK cells displayed basolateral localization, but does not require the same region as TβRII. A schematic shows the features of Punt relative to the plasma membrane (E). The amino-terminal ectoplasmic portion is separated from the cytoplasmic domain by the vertical bar indicating the plasma membrane. The HA tag and the corresponding position of the LTA motif are shown in red, with numbers indicating FL Punt and truncation positions. (F–I) xz projections from confocal images of intact epithelium transiently expressing HA-Punt constructs, with apical to the top of each image, as indicated. Nuclei are stained with DAPI (blue) and surface receptors are stained with anti-HA (red). FL Punt and variants harboring deletions of extended segments of the cytoplasmic domain were all detected at the basolateral membrane but excluded from the apical membrane. See S2 Fig for additional deletion constructs that retain basolateral distribution. Scale bars: 100 μm for B, C, D; 15 μm for B′, C′, D′; 25 μm for F–I (scale varies slightly for xz projections in F–I). BLT, basolateral targeting; FL, full length; MDCK, Madin-Darby canine kidney; TM, transmembrane region.

## Identification of the BLT motif in the juxtamembrane region of Punt

The combined results from the cytoplasmic truncations in MDCK cells with the chimeric receptors in wing discs indicate that the cytoplasmic juxtamembrane portion of Punt harbors a basolateral determinant. We examined the amino acids between the chimeric protein junction and the end of the smallest truncation for predicted targeting motifs and evolutionary conservation. No canonical BLT motif [23] is present in this sequence. There is no detectable amino acid conservation in this region among the complete set of *Drosophila* TGF-β receptors (3 Type I and 2 Type II) [24]. However, there is strong conservation of several positions within many insect Punt homologs (Fig 3A). These amino acids fall between the transmembrane motif and the kinase domain, a region not linked to the core ligand binding and signal transduction functions of the receptors.

To directly test the requirement of this region in basolateral localization, we deleted 10 or 19 amino acids and determined the membrane targeting of the altered proteins. The 10 amino

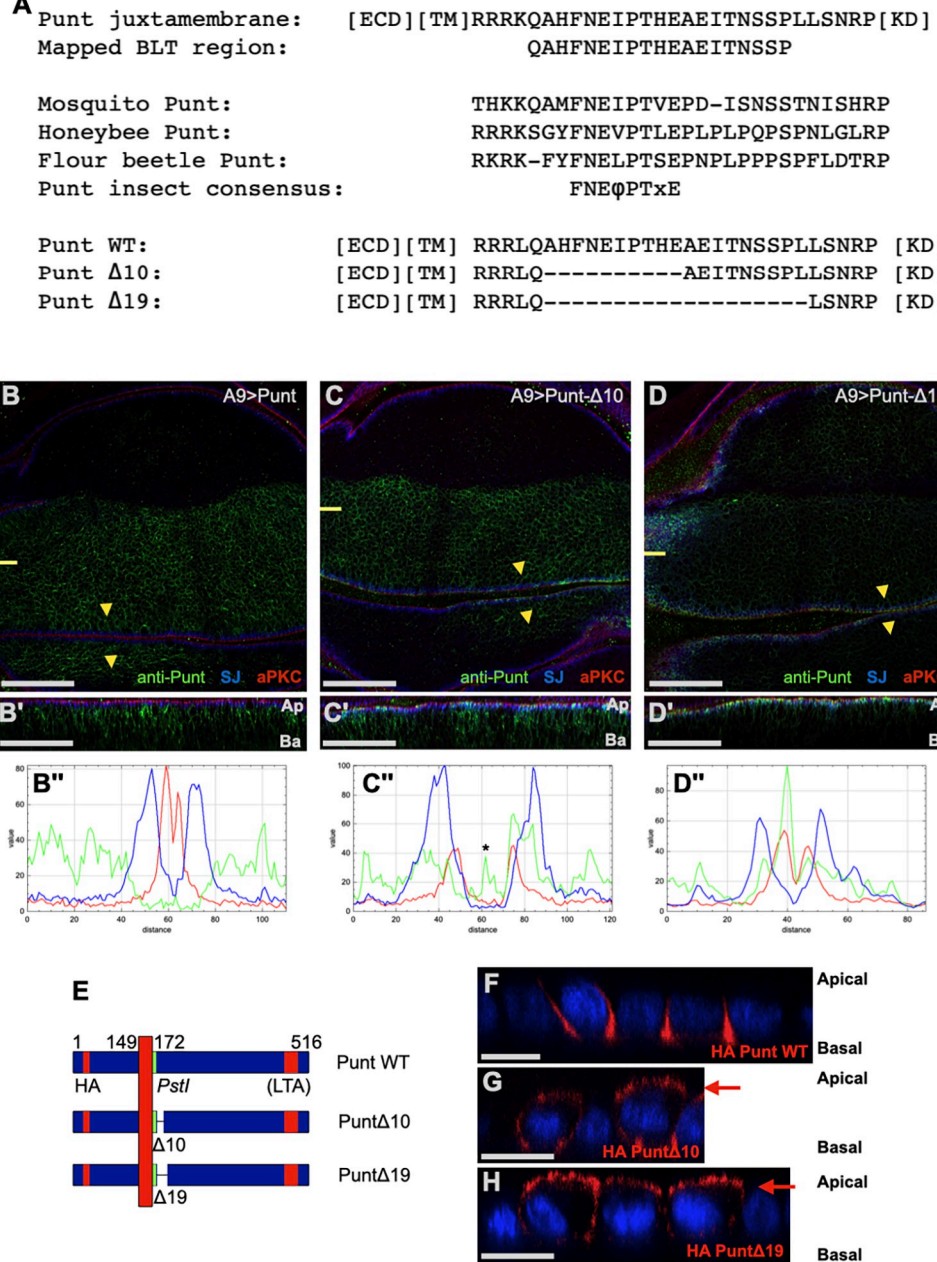

**Fig 3. The cytoplasmic juxtamembrane region of Punt harbors the BLT determinant. (A)** The juxtamembrane region of Punt in the context of the ECD, TM, and KD [25]. The BLT region, as delimited by behavior of chimeric and truncation constructs, is indicated. Amino acid sequence alignment of several insect Punt homologs revealed a core conserved sequence between the predicted transmembrane helix and the KD. The sequence of internal deletions (Δ10 and Δ19) relative to Punt WT is shown; all constructs in this series harbor a K to L point mutation. **(B–D)** Deletion of the BLT degrades the basolateral restriction of Punt protein expressed in wing disc epithelia with A9-GAL4, detected with anti-Punt IF. Single prime (′) images are xz projections at the plane indicated by the short yellow lines. Yellow arrowheads indicate position and direction of RGB profile shown in double prime (″) images; the asterisk in C″ indicates signal outside the cell layer that does not correspond to surface protein. **(E–H)** Deletion of the BLT also destroyed basolateral restriction in MDCK cells. Punt was detected mainly in the lateral membrane region (F), whereas Punt-Δ10 and Punt-Δ19 displayed robust apical staining (level marked by red arrows) in addition to basolateral staining (G, H). DAPI staining labels nuclei (blue) and receptor was visualized by anti-HA IF (red). Scale bars: 50 μm for B-D, B′-D′; 25 μm F-H. See S3 Fig for evidence of activity of internal deletion proteins and S4 Fig for corroborating results using a different GAL4 driver and epitope antibody. BLT, basolateral targeting; ECD, extracellular domain; KD, kinase domain; MDCK, Madin-Darby canine kidney; TM, transmembrane region; WT, wild-type.

acid deletion removes the insect-conserved residues, and the 19 amino acid deletion additionally removes the remaining residues present in the shortest truncation (Fig 3A). These internal deletions did not destroy signal transduction activity of Punt as judged by accumulation of ectopic p-Mad (the phosphorylated active form of Mad) in wing discs (S3 Fig). Compared to wild-type (WT) Punt, the deletion proteins displayed apical staining, with the 19 amino acid deletion causing more severe apicalization than the 10 amino acid deletion (Fig 3B–3D). Similar results were observed with an additional GAL4 driver and staining for another epitope (S4 Fig). Strikingly, parallel behavior was observed in MDCK cells: The removal of 10 amino acids led to clear apical staining, and Δ19 had somewhat stronger apical localization (Fig 3E–3H). The prediction that a targeting determinant resides in the cytoplasmic juxtamembrane portion of Punt was borne out by these results, and we coin this sequence the BLT determinant of Punt.

To confirm this result in a context preserving the spacing of the intracellular portion of Punt relative to the transmembrane domain, we tested the behavior of 2 variants of fruit fly Punt harboring juxtamembrane sequences from related proteins. We used the corresponding portion of honeybee Punt protein as one donor to test the functional conservation of this sequence in insects (Fig 4A). Transgenic flies expressing the Punt[Apis] protein showed predominantly basolateral staining in wing discs, indicating functional conservation among insect Punt proteins (Fig 4B and 4C). Because Wit is apically enriched, we used the *Drosophila* Wit juxtamembrane sequence to test if this amino acid sequence restricts membrane location. Punt [Wit] showed apical enrichment, much like full length Wit (Fig 4D). This strong effect of juxtamembrane sequences on protein distribution indicates that BLT is a conserved feature of insect Punt juxtamembrane sequences.

To further probe the structure-function aspects of the BLT motif at the level of amino acid sequence, we characterized a series of BLT point mutations targeting the amino acids conserved in insects. Alanine substitutions were made for 3 positions at a time and the localization of the resulting protein was determined in wing discs. The FNE, EIP, and PTE variants (named for mutated conserved residues) were detected primarily at basolateral membrane regions (Fig 4E, S5 Fig), similar to the WT starting protein. To specifically address a potential role for charged side chains, an EEE variant was tested, which was also basolateral (S5 Fig). More extensive mutation including 7 conserved residues (FNEIPTE) led to mixed apical and basolateral distribution, and a more drastic alteration mutating the entire 19 amino acid stretch covering the minimal BLT mapped by the truncation series led to strong apical localization (Fig 4F and 4G), which resembled the Δ19 and Punt[Wit] proteins. Considering these results together, we conclude that the BLT amino acid sequence is robust in the face of mutation and that both the section containing conserved amino acids and the surrounding sequence contribute to the BLT of Punt.

## The Punt BLT motif is a dominant, transferable targeting signal

Having established a requirement of the Punt juxtamembrane region in basolateral restriction, we conducted swap experiments to address sufficiency by asking if the BLT acts as a dominant BLT motif. Wit shares many signaling properties with Punt, but as described above showed a starkly different localization in wing disc epithelium. We replaced the corresponding portion of Wit with the Punt BLT and assessed membrane localization (Fig 4A). A control version of Wit (with a single amino acid change from a cloning scar) displayed strong apical staining that overlaps with the aPKC marker (Fig 4I). Wit[Punt] behaved like intact Punt, with nearly all staining excluded from the junction and apical membranes (Fig 4J, S6 Fig). The Punt BLT motif is thus a dominant moveable BLT motif that confers basolateral restriction in wing disc

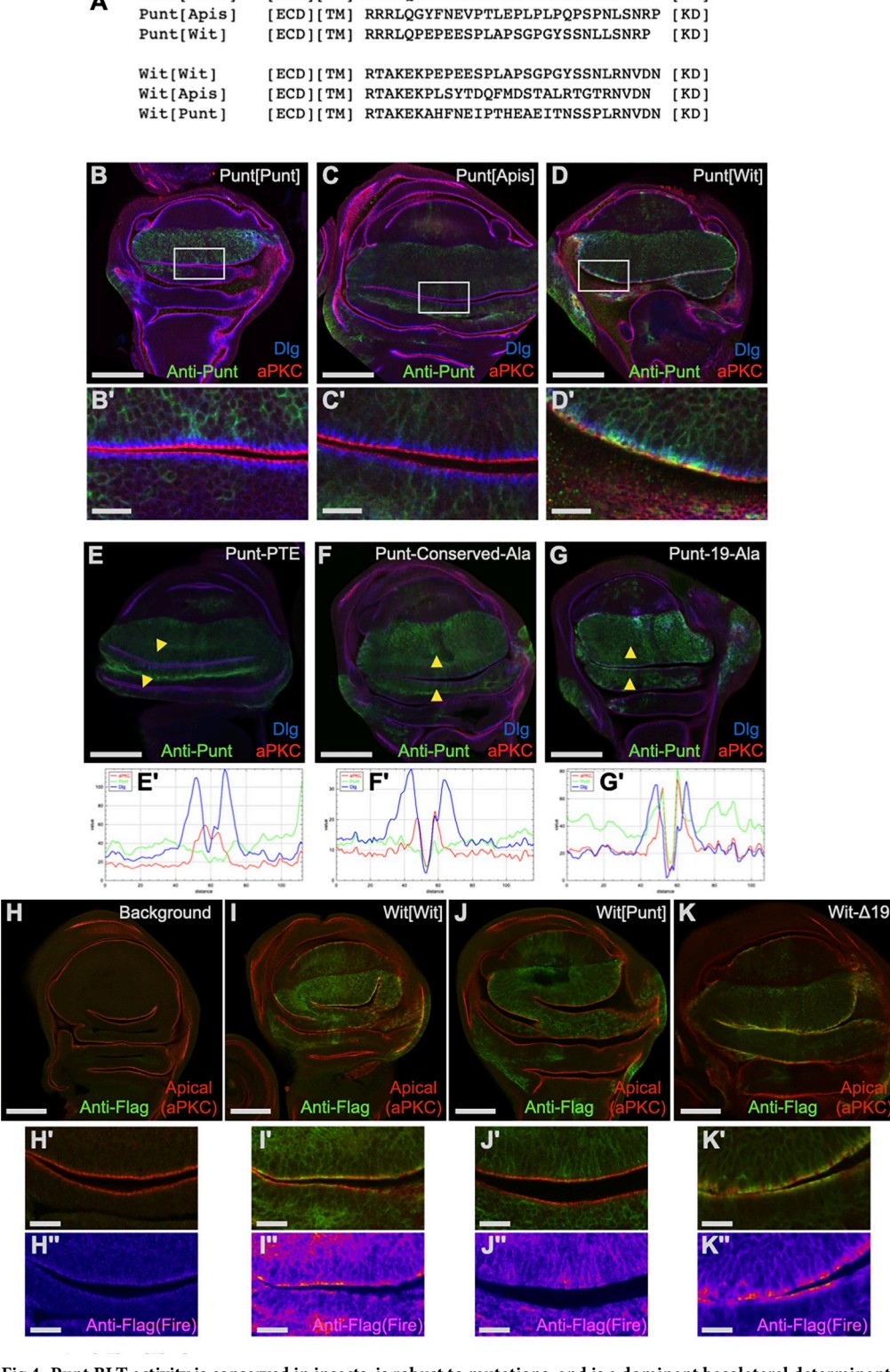

**Fig 4. Punt BLT activity is conserved in insects, is robust to mutations, and is a dominant basolateral determinant in the wing disc.** (**A**) Amino acid sequences of the juxtamembrane substitution constructs. The backbone protein is indicated with the incorporated BLT region indicated in brackets. Punt[Apis] and Wit[Apis] harbor juxtamembrane sequences from the respective honeybee Type II receptor ortholog. (**B–D**) BLT function is conserved in insect Punt proteins. A9-Gal4 driving the control UAS-Punt[Punt] protein revealed the absence of staining at or apical to the

junctions (B). The corresponding sequence from honeybee also supported basolateral restriction of the Punt[Apis] protein (C). The corresponding Wit sequence substituted for Punt's failed to provide BLT activity, with the Punt[Wit] protein detected in a pattern resembling Wit-GFP (D). See S5 Fig for 3-color staining profiles corresponding to B–D. **(E–G)** Mutation of the BLT leads to varied apicalization depending on the severity of the mutation. Representative staining and signal profiles for mutant versions of Punt with increasing number of changed amino acids expressed with A9-Gal4 are shown. The Punt-PTE variant generally showed a lack of staining in the apical domain (E and E′), similar to WT Punt. Mutation of the insect-conserved residues to Alanines (Punt-Conserved-Ala) led to moderate and spatially varied overlap with the aPKC apical marker (F and F′). Mutation of the entire 19 amino acids of the BLT region (Punt-19-Ala) abolished basolateral enrichment as shown by the overlap of Punt staining with Dlg and aPKC (G and G′). **(H–J)** Moving the Punt BLT to the juxtamembrane position of Wit caused basolateral restriction. Flag IF gives a nontrivial background signal enriched at the apical membrane (H). The control Wit[Wit] protein driven by A9-Gal4 showed enrichment at the apical membrane (I). Wit[Punt] displayed primarily basolateral distribution with many apical regions showing staining at background levels (J). A Wit-Δ19 construct had similar distribution to Wit, indicating that the juxtamembrane region of Wit does not control the apicobasal distribution (K). Double prime (″) images are Fire false-color displays of the anti-Flag channel. See S6 Fig for profiles for images corresponding to H-K. Scale bars: 100 μm for primary images; 15 μm for prime and double prime microscope images. BLT, basolateral targeting; WT, wild-type.

epithelia on an otherwise apically enriched protein. Deletion of the juxtamembrane region of Wit did not change the apicobasal protein distribution (Fig 4K, S6 Fig), indicating that the altered distribution of Punt[Wit] is due to the Punt BLT sequence and not the loss of the corresponding region of Wit. Further, we queried the positional requirement of the BLT by appending it to the carboxyl terminus of an otherwise apicalized form of Punt and assaying epithelial distribution. Punt-Δ19-CBLT retains the apical staining of Punt-Δ19 (S5 Fig), indicating that the motif does not function at the carboxyl terminus. Since it functions in a juxtamembrane position but not at the carboxyl terminus, we conclude that the function of the BLT motif as a dominant *cis*-acting determinant is contingent on topology.

## Punt activity is regulated by polarized membrane localization

All variants of Punt used for our localization tests were active when overexpressed in the wing (S3 Fig). This may mask functional differences since any overexpressed Type II receptor can bypass normal controls to generate ectopic signaling. To correlate the localization of Type II receptors with function under controlled expression conditions, we utilized transgenic lines with a set of UAS constructs recombined into fixed genomic attP sites. For Wit, we tested 3 constructs for their ability to rescue *wit* mutants when expressed in neurons with *elav*-GAL4, as has previously been shown for Wit [18]. The control protein, Wit-FLAG, provided full rescue to viability in this assay (S7 Fig). Wit[Apis] and Wit[Punt] also provided full or substantial rescue (S7 Fig), indicating that the juxtamembrane domain of Wit is not critical for its function in this assay and that the Punt BLT does not grossly interfere with Wit function in neurons. We also tested the ability of Punt variants to rescue *wit* mutants. Any form of Punt overexpressed pan-neuronally supported rescue of otherwise lethal *wit* mutants (S7 Fig). This indicates that some Type II activity is sufficient in neurons to support adult viability, but yields no evidence that the BLT region is critical to regulate the activity.

A parallel attempt to rescue *put* mutants with UAS-controlled proteins was hampered by toxicity of overexpressed Punt. Widespread ectopic expression of Punt was lethal at 25 degrees in an otherwise WT background or a *put* mutant background (S7 Fig). The *put* alleles we used for the rescue tests are temperature sensitive, showing variable survival at lower temperatures [26], thus precluding assessment of relative rescue activities with reduced UAS expression levels at lower temperatures. However, we did leverage the temperature sensitivity of the GAL4 system to uncover differential activity of overexpressed Punt variants with altered BLT regions. Overexpressed Punt[Wit] produced a lower degree of lethality than Punt[Punt] or Punt[Apis] over a range of temperatures with the spatially restricted *en*-GAL4 driver (Fig 5A). Similar

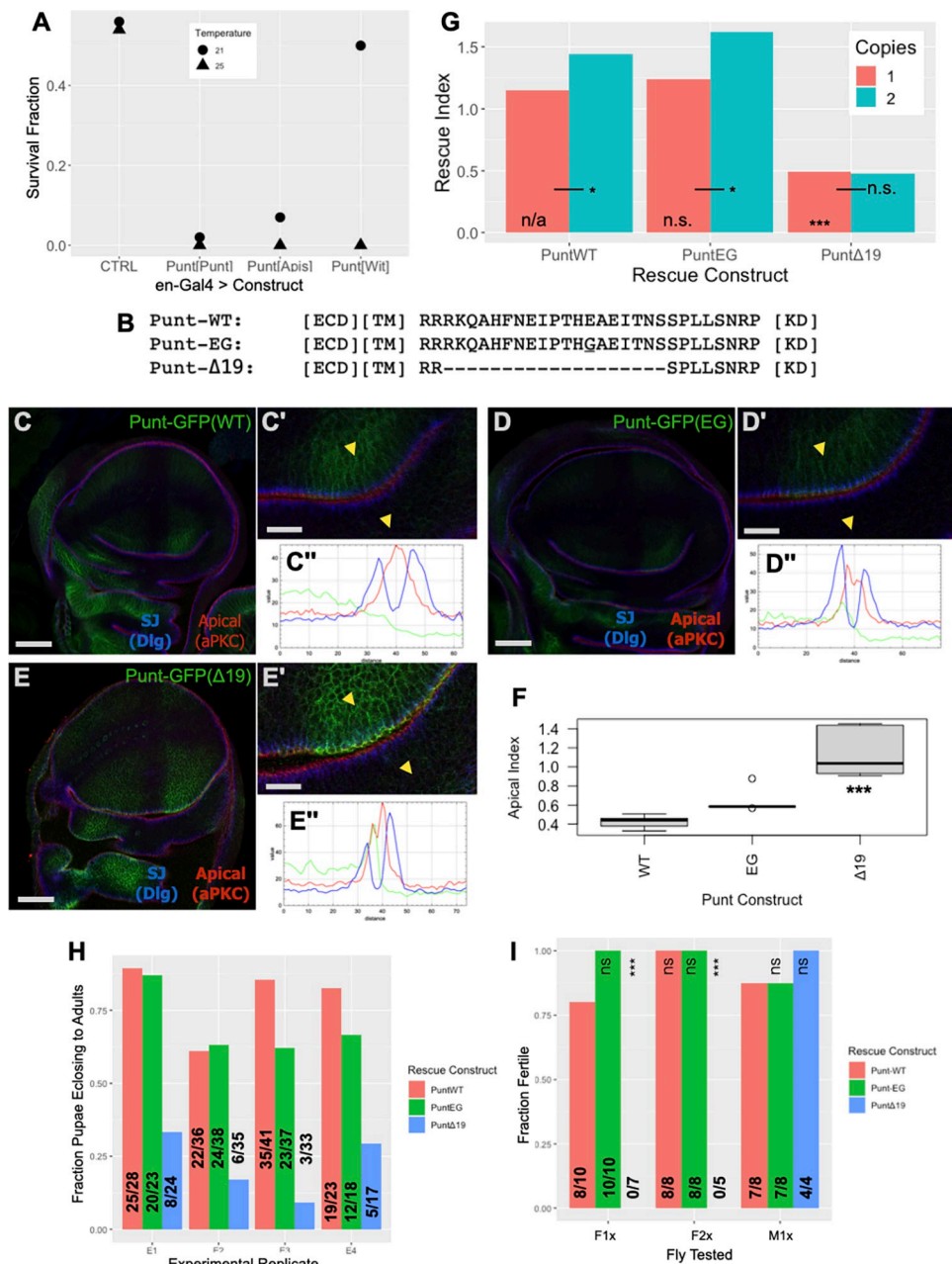

**Fig 5. The BLT region influences the activity of overexpressed Punt and is required for animal fitness in a rescue assay. (A)** BLT-containing Punt proteins caused more ectopic lethality than a form of Punt that is apicalized in imaginal discs. *en*-GAL4 driving UAS-Punt[BLT] proteins revealed differential temperature sensitivity for lethality; Survival Fraction of 0.5 is full survival relative to control balancer chromosome (A). Animals counted at 21 degrees: control, 605; Punt[Punt], 465; Punt[Apis], 423; Punt[Wit], 484. At 25 degrees: control, 390; Punt[Punt], 216; Punt [Apis], 284; Punt[Wit], 82. See S7 Fig for additional overexpression results. **(B–G)** Genomic rescue constructs (transgenes composed of a 7 kb genomic region with a carboxyl-terminal GFP tag) link viability to presence of the BLT. Juxtamembrane amino acid sequences for the rescue constructs (B). Punt-GFP staining by anti-GFP IF showed primarily basolateral restriction (C). The EG point mutant displayed mild apicalization (D), whereas the Δ19 construct presented a stronger apical signal (E). Single prime (′) panels show zoomed image with yellow arrowheads indicating the profiles in double prime (″) panels. The Δ19 version of Punt showed more junction and apical staining (Apical Index) than the WT and EG versions (F); statistical code is for comparison of Δ19 and WT constructs. See S8 Fig for expression pattern of the rescue construct-expressed Punt proteins. PuntWT and EG constructs provide sufficient activity to support robust survival to adulthood, but the Δ19 construct supported only limited survival; Rescue Index of 1.0 is full survival relative to control balancer chromosome genotype (G). Animals counted for one copy rescue: WT,

585; EG, 564; Δ19, 589 and for 2 copy rescue: WT, 231; EG, 197; Δ19, 251 (G). Statistical codes for one copy of PuntEG and PuntΔ19 are in comparison to one copy of PuntWT; those for 2 copy tests are relative to the matching one copy tests. Lethal phase observations showed that Punt-GFPΔ19 animals exhibited increased pupal lethality in each of 4 biological replicates E1-E4 (H); numerical data shown on graph. WT and EG rescue constructs supported female fertility, but the Δ19 construct did not (I); F1x represents Females with 1 copy of the rescue construct, F2x represents Females with 2 copies, and M1x represents Males with 1 copy; numerical data and statistical significance shown on graph. Scale bars: 100 μm for C–E, 15 μm for C′–E′. n.s. indicates $p > 0.0.5$, * indicates $p < 0.05$, *** indicates $p < 0.0001$ for Fisher exact tests (G, I) or TukeyHSD (F). The data underlying all the graphs shown in the figure can be found in S1 Data. BLT, basolateral targeting; WT, wild-type.

results were obtained for gender-biased lethality with the X-linked A9-GAL4 driver, which is more active in males than females (S7 Fig).

## *In vivo* rescue efficiency depends on the BLT motif

To bypass the toxicity of overexpressed Punt, we generated transgenic lines encoding Punt variants expressed from endogenous regulatory elements. A large BAC-based construct expressing GFP-tagged Punt (Punt-GFP) provided full rescue of *punt*$^{135/P1}$ animals at 25 degrees (S8 Fig). A smaller construct including the *punt* genomic region was also able to provide genetic function, with equivalent activity for Punt and Punt-GFP version. Survival, fertility, and overt behavior were fully rescued by this construct. To correlate localization with function, we compared the WT rescue construct to a BLT point mutant (EG) and a deletion affecting the BLT (Δ19) (Fig 5B). As shown by anti-GFP immunofluorescence (IF), the WT construct was basolaterally restricted in the wing disc epithelia (Fig 5C), in line with the distribution of overexpressed Punt. The EG variant showed a limited degree of increased apical staining, and the Δ19 protein was more strongly enriched apically (Fig 5C–5F). In this context, the EG construct provided rescue activity equivalent to the WT construct. Deletion of the BLT domain, however, severely reduced the rescue activity of Punt as assayed by survival to eclosed adults (Fig 5G). Lethal phase analysis showed that animals with Punt-Δ19 as the sole source of Punt frequently failed during pupation (Fig 5H). Additionally, eclosed adults displayed reduced movement and died within several days. Finally, the surviving females were completely sterile (Fig 5I). Taken together, these results reveal that the BLT portion of Punt is crucial for full viability and fertility.

Having established the requirement of the Punt BLT for overall fitness, we returned to the wing to probe the signaling consequence of altering protein localization. We considered 2 general outcomes: that apical Punt would create a gain-of-function situation due to exposure to and activation by the lumenal Dpp pool or that redirection of Punt from basolateral to apical membranes would create a loss-of-function situation for Dpp signaling. To distinguish between these outcomes, we analyzed molecular and morphological features associated with the Dpp morphogen gradient. In an overexpression context, the lethality results described above for UAS-driven Punt variants are consistent with basolateral Punt having more toxicity, which could suggest this position supports more signaling activity. We linked this to signal transduction by examining p-Mad levels upon expression of Punt variants with the *en*-GAL4 driver. Based on stronger p-Mad staining in the posterior compartment, the basolateral proteins Punt[Punt] and Punt[Apis] produce more ectopic signaling than the apicalized Punt [Wit] version (Fig 6A–6E).

To address the same question under physiological conditions, we stained rescued animals for p-Mad as a direct readout of BMP signaling downstream of endogenous Dpp. Larvae with a single copy of Punt-WT had a stereotypical p-Mad pattern and overtly normal wing discs (Fig 6F). The corresponding EG point mutant genotype also had normal p-Mad detection

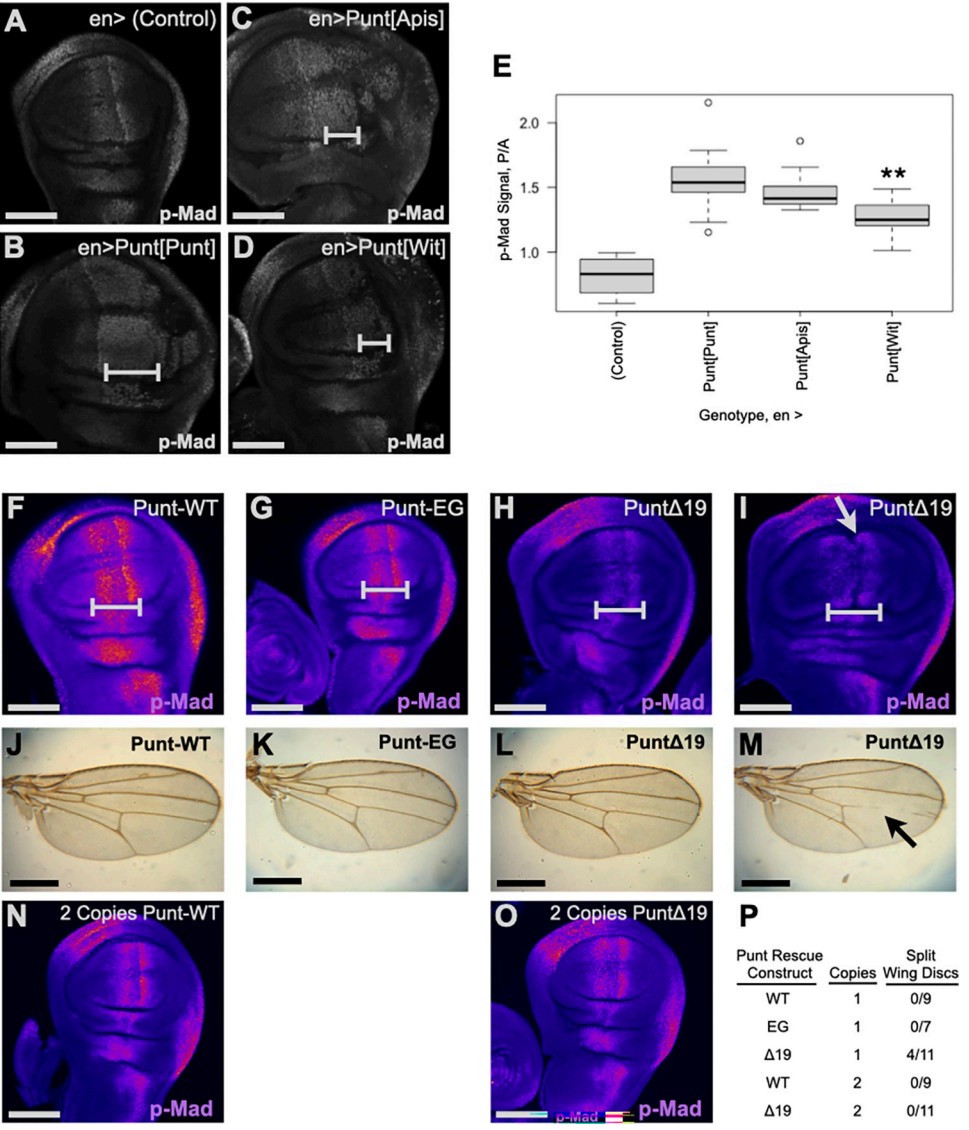

**Fig 6. The BLT motif positively supports Dpp signaling. (A–E)** Ectopic activity of overexpressed Punt was diminished upon apicalization. p-Mad pattern in the wing disc for control (A) and *en*>Punt[BLT] proteins (B–D). Ectopic p-Mad staining is marked by brackets (B–D). The ratio of p-Mad signal from the posterior compartment (*en*-positive) versus anterior compartment increased for each Punt[BLT] protein, but less so for Punt[Wit] (** $p < 0.01$ Punt[Wit] versus Punt[Punt]) (E). **(F–M)** Apicalized Punt did not fully support Dpp signaling or wing development at a dosage of one copy. Representative wing disc p-Mad staining from *put* mutant larvae rescued by the indicated construct (F–I), shown as false color Fire scale of MIPs. Note reduced p-Mad intensity in the pouch region above the bracket in panels H-I. Δ19 animals showed a variable morphology, with 4 of 11 discs having an abnormal furrow along the A/P boundary (gray arrow in panel I). **(J–M)** Wings from adults from rescued genotypes. Δ19 wings (L, M) showed variability in vein patterns (arrow in M). **(N, P)** Signaling and tissue patterning is dose dependent. Two copies of the Δ19 rescue construct boosted Dpp signaling to normal levels (N, O). One-copy images (F–I) are from one staining batch and 2-copy images (N, O) are from a different staining batch. Developmental defects were only observed with the Δ19 construct at a dosage of one copy (P). Scale bars: 100 μm A–D, F–I, N, O; 500 μm for J–M. The data underlying the graph shown in this figure can be found in S1 Data. BLT, basolateral targeting; Dpp, Decapentaplegic.

within normally shaped discs (Fig 6G). The Δ19 rescue animals, however, frequently displayed a reduced p-Mad staining intensity, and a fraction of the discs presented an abnormal split morphology (Fig 6H and 6I). Both of these phenotypes are consistent with reduced Dpp

signaling [9,11]. These results indicate that a Dpp morphogen gradient can form without the membrane targeting function of Punt's BLT, but the output is weakened and exposes the tissue to developmental error. Wings from the subset of surviving adults correspondingly displayed variable shape and veination defects compared to the fully rescued genotypes (Fig 6L and 6M). To test the hypothesis that the developmental defects observed with one copy of the ΔBLT construct are due to limiting amounts of basolateral Punt, we examined wing discs with 2 copies of the rescue construct. At this dosage, the Δ19 variant fully supported wing disc patterning (Fig 6N–6P), but did not boost viability or female fertility (Fig 5G and 5I).

## Tissue-specific utilization of the BLT and *trans*-acting sorting factors

Taken together, the overexpression and endogenous level activity assays indicate that altering the localization of Punt changes its activity in at least 1 tissue, but it is not strictly required for signaling. We were intrigued that the activity of Punt in the wing tracked with overall animal survival. To define the generality of the localization program, we determined the localization of Punt and Wit in 2 other epithelial tissues. In larval salivary glands, both Punt and Wit were detected only in basolateral membrane regions but were excluded from junctions (Fig 7A and 7B). In follicle cells of the egg chamber, both Punt and Wit were indiscriminately distributed basolaterally and apically (Fig 7C and 7D). Thus, in 3 different fly epithelia, we observed 3 distinct distribution profiles, indicating that for a given receptor, membrane targeting schemes vary with tissue.

The conclusion that the Punt BLT confers basolateral restriction in the wing disc epithelium is based on observation of stable protein distribution. Generically, this preferential localization can be achieved by specific delivery to basolateral surfaces or by selective removal from apical surfaces or by a combination of these actions. As a broad test of these mechanisms, we disrupted a group of trafficking factors to ascertain their requirement for Punt basolateral restriction. We induced RNAi in clones to knock down candidate genes in animals expressing Punt-GFP from a rescuing transgene. This approach is advantageous because it precludes any overexpression or ectopic signaling effects of Punt, avoids secondary consequences of loss of cell biology functions in the entire disc, and permits direct comparison of neighboring cells within the same tissue, thus avoiding potential changes from developmental stage, environmental conditions, or detection parameters.

Adaptor protein (AP) complexes are key mediators of vesicular sorting and membrane trafficking. The *Drosophila* genome encodes subunits for 3 AP complexes [27]. AP-1 has been implicated in sorting mechanisms that direct cargo to restricted membrane domains, whereas AP-2 has been linked to endocytosis and early endosome movement [28,29]. We tested RNAi lines for genes encoding *Drosophila* AP-1 and AP-2 components. RNAi constructs for AP-1/2β, AP-1γ, AP-2α, AP-2σ, and AP-2μ were active as judged by lethality upon expression with a constitutive GAL4 driver (see Methods). Wing disc clones expressing each RNAi element were generated by hsFlp, and the localization of Punt-GFP was compared in clones and nearby normal cells. In nearly all AP-1/2β knockdown clones, Punt-GFP was detected at the apical membrane at levels higher than surrounding cells (Fig 7E). Because the *Drosophila* β1/2 subunit (Bap) is thought to assemble into both AP-1 and AP-2 complexes, we tested AP-1γ to clarify the requirement of AP-1. AP-1γ knockdown clones also have apical Punt (Fig 7F). We thus conclude that the AP-1 complex is required to produce or maintain the basolateral distribution of Punt.

To test the requirement of endocytosis routes, we knocked down several AP-2–specific subunit genes. In each case, Punt-GFP distribution was not altered in the clones and there was no apparent change in level (Fig 7G, S9 Fig). AP-2 has been shown to affect endocytosis in several

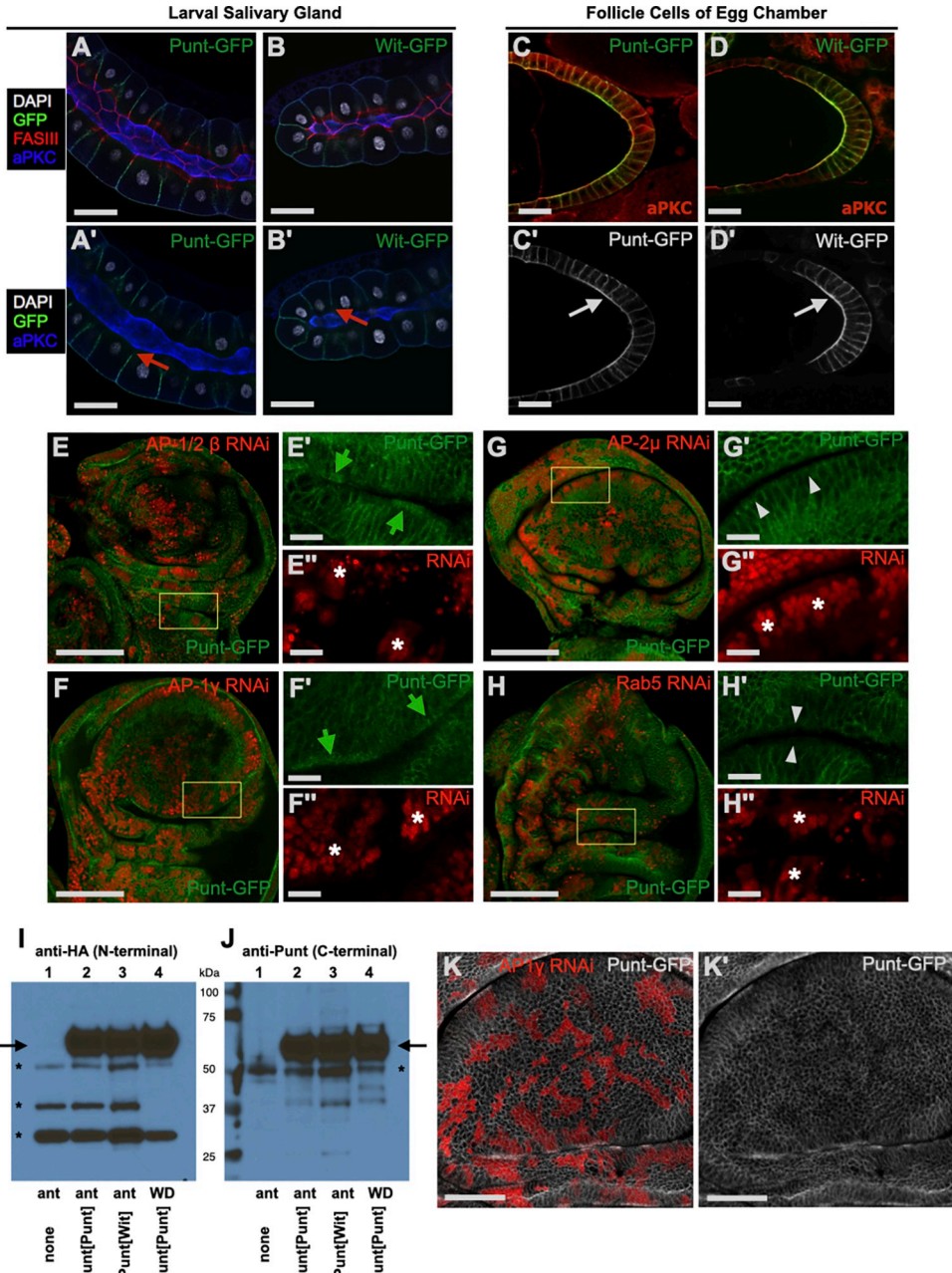

**Fig 7. Membrane distribution of receptors varies by tissue and requires AP-1 sorting machinery in the wing disc.**
**(A, B)** Punt-GFP and Wit-GFP displayed basolateral restriction in the larval salivary gland epithelium. Confocal sections through the center of the salivary gland show cell polarity pointing exterior (basal) to interior (apical) with a central lumen (aPKC). Red arrows point to gap in GFP fluorescence signal at the SJ (A′, B′). **(C, D)** Punt-GFP and Wit-GFP showed general membrane presentation in the follicle cell epithelium in the ovary. Arrows point to unambiguous GFP signal at the apical surface (C′, D′). **(E–H)** Screening for apicalization of Punt-GFP in RNAi clones for trafficking factors. Clones are marked in red in whole disc confocal image and double-prime insets. GFP IF signal is shown in isolation in single-prime insets. Asterisks in the red channel point to example RNAi clones (E″–H″). Green arrows point toward the apical surface and mark ectopic apical staining in RNAi clones relative to neighboring WT cells (E′, F′). White arrowheads point toward the apical membrane for conditions where the apical level of Punt-GFP did not change between RNAi clones and control cells (G′, H′). Punt-GFP is observed at the apical membrane near marked clones with RNAi knockdown of AP-1/2β (E) or AP-1γ (F). No such apical staining was observed in clones with RNAi knockdown for AP-2 components such as AP-2μ (G) or for Rab5 (H). **(I–J)** Western blot detection of Punt [Punt] and Punt[Wit]. Samples were collected from *nub*-GAL4 driving the indicated Punt construct. "ant" is anterior half of the larvae, including discs, and "WD" is specific dissection of Wing Discs. The HA epitope near the N-terminus

did not detect significant specific degradation products (I). Detection of the Punt antibody epitope near the carboxyl terminus also did not reveal any significant specific bands that would suggest clipping (J). Mobility markers are indicated by kDa labels, arrows indicate expected position of Punt, and asterisks mark background bands not attributed to specific Punt detection. Punt[Wit] level was not increased with either antibody (I, J). **(K)** AP-1γ knockdown leads to a reduction in basolateral Punt. A single confocal plane through the lateral portion of the columnar cells of the wing pouch with AP-1γ RNAi clones marked in red (K). Multiple areas with reduced Punt-GFP staining near the center of the pouch correspond to the RNAi clones (K′). The streaks of increased Punt-GFP staining in panels K and K′ correspond to the apical sides of RNAi clones. Scale bars: 100 μm for A, B, E, F, G, H; 50 μm for K, K′; 25 μm for C, D; 15 μm for E′–G′, E″–G″. See S9 Fig for a summary of the data supporting the sorting model. AP, adaptor protein; RNAi, RNA interference; WT, wild-type.

cell types, including the wing disc epithelium [30], but the relative contribution of AP-2 to endocytic flux is unclear. As a more stringent challenge to endocytosis, we tested knockdown clones of Rab5, which is required for early endosome formation in *Drosophila*. As previously observed [31], most discs were devoid of Rab5 RNAi clones, but some discs harbored a large proportion of clone cells. The morphology of the disc was grossly abnormal, but epithelial polarity was maintained and Punt-GFP remained in membrane domains basal to the junctions (Fig 7H). We thus found no evidence that endocytosis plays a significant role in restricting Punt's membrane distribution. Combining the AP-1 data with the negative results for AP-2 and Rab5, we conclude that AP-1–dependent selective delivery is the most likely mechanism generating the basolateral distribution of Punt via its BLT motif.

Another potential mechanism to remove Punt from the apical membrane is proteolytic cleavage, as has been observed for other receptors including mammalian TGF-β receptors [32]. We analyzed Punt[Punt] and Punt[Wit] expressed in the wing disc by western blot. We observed no difference in protein level nor detected any shorter peptide containing either the amino-terminal HA epitope or the carboxyl-terminal Punt antigen that differed between Punt[Punt] and Punt[Wit] (Fig 7I and 7J). If BLT activity controls Punt localization without altering levels, it follows that apicalization of Punt leads to reduced basolateral Punt levels. Indeed, clones with impaired AP-1 activity showed reduced basolateral Punt (Fig 7K), supporting the model that the basolateral level of Punt is critical for robust Dpp signaling and subsequent tissue growth and patterning.

## Discussion

### The BLT of Punt directs restricted basolateral localization in the wing disc and MDCK cells

The goal of this study was to determine if polarized membrane localization of TGF-β pathway receptors impacts growth and patterning of the wing disc. First, we determined the membrane distribution of receptors that mediate Dpp signaling. Tkv, the primary BMP pathway Type I receptor, was detected in both basolateral and apical membrane domains, with apparent enrichment at the cellular junctions, consistent with studies showing Tkv protein in apical cytoneme projections [33]. In contrast, the primary Type II receptor Punt was detected in the basolateral membranes in the wing disc epithelium, but excluded from the apical membrane and cell–cell junctions. Another Type II receptor, Wit, was enriched apically but also detected in basolateral domains. The varied membrane distribution patterns among these receptors shows that basolateral localization is not a global feature of TGF-β pathway receptors. We focused on Punt because its restricted localization could influence Dpp signaling.

To what extent is membrane targeting conserved between insect and mammalian epithelia? The basolateral restriction of TβRII and Punt prompted us to assess evolutionary conservation. Notably, Punt exhibited basolateral restriction in MDCK cells. Truncation and chimeric

constructs pointed to a short juxtamembrane region of the Punt cytodomain, the BLT. Notably, this region is distinct from the LTA motif, which is required for TβRII restriction [20]. Deletion of the BLT segment abolished basolateral restriction in both MDCK cells and wing disc epithelial cells, establishing the necessity of this portion of the receptor. Insertion of the Punt BLT into Wit switched its localization to basolaterally restricted, indicating sufficiency.

None of the several previously described flavors of BLT motifs [29] is present in the short active region of Punt. However, sequence alignments revealed a core FNEφPTxE sequence within the BLT region among several Punt insect homologs, suggestive of a sequence-specific element. Simultaneous mutation of several sets of conserved residues did not abolish the basolateral preference of overexpressed Punt, indicating robustness of the BLT region for BLT. In fact, the only 2 mutations that completely erased the basolateral preference were 19-Ala and Punt[Wit]. On the other hand, a single E-to-G mutation within the conserved sequence led to partial apicalization of Punt-GFP expressed at the endogenous level. There may be a biochemical or structural difference between the E-to-G and alanine substitution included in the PTE variant, or there may be differences in the trafficking flux of endogenous and overexpressed Punt. In addition to the intermediate behavior of Punt[conserved], we uncovered other evidence that nonconserved residues contribute to BLT function. The Δ19 protein displayed more apicalization than the Δ10 version. The robust basolateral sorting of the Punt[Apis] protein is interesting in the context of amino acid conservation. If viewed as a mutant of all the nonconserved residues, this would suggest that specific residues at nonconserved positions are not important. However, since the 19 amino acid stretch is from a native protein, the BLT from honeybee Punt has likely been under functional selection to support BLT activity. Collectively, our results support several conclusions about topology. The failure of a carboxyl-terminal BLT sequence to confer basolateral restriction suggests that the BLT associates with factors near the membrane. Deletion of 10 amino acids containing the conserved residues had a much stronger effect on localization than alanine mutations, suggesting that local spacing is important for BLT activity. Further work, including identifying binding proteins as discussed below, is required to explain the mechanism of the novel basolateral sorting signal embedded in the BLT.

Mapping studies of 3 TGF-β/BMP receptors with basolateral restriction have uncovered 3 different targeting motifs, none of which corresponds to canonical motifs (this work and [20,34]). Direct mapping approaches can identify *cis* determinants of restricted apicobasal expression, such as a BLT sequence in the cytoplasmic of TβRIII that can be disabled by mutation of a single Proline residue [7]. In the context of myriad families of transmembrane proteins, all of which have the potential for regulated apicobasal presentation, we project that there is a substantial pool of heretofore unrecognized sorting motifs.

## Relevance to Dpp signaling in the wing disc

Armed with the identification of the *cis*-acting BLT motif of Punt, we assessed its role in regulating signaling by blocking its function and monitoring signaling *in vivo*. This approach requires direct study of each protein of interest to identify and modify the *cis*-acting sorting determinants. An alternative approach has been developed to assess the outcome of coercing proteins to different apicobasal membrane domains [35]. That approach offers a higher throughput platform for exploring targeting-function relationships for a variety of membrane proteins, but will not directly shed light on the *cis*- and *trans*-acting machinery that execute the normal sorting. For the case of Punt in the wing disc, we wished to define both the consequence of mis-localization and the mechanism of the specific BLT.

At the organismal level, removal of the BLT region severely harms fitness. WT *punt* rescue constructs restored full viability and fertility, but a construct lacking the BLT provided only

partial viability and rendered surviving females sterile. Multiple functional domains of TGF-β receptors have been characterized [36], including the ectoplasmic ligand-binding domain, cytoplasmic GS box (found in Type I receptors), and cytoplasmic kinase domain. However, function has traditionally not been assigned to the cytoplasmic juxtamembrane portion. Notably, an exception is the VxxEED juxtamembrane motif that acts as a basolateral sorting motif for TβRI [34]. This portion of the receptor thus appears to offer a target for natural selection that can regulate receptor function outside the core ligand-binding and kinase domains. Our data show that for Punt, one regulatory function is to control polarized receptor presentation in epithelia. It is possible that additional regulatory functions are embedded in this region, perhaps involving the insect-conserved amino acids. In particular, we have not linked the reduced survival nor sterility observed in animals lacking the BLT with BLT. Additionally, we note that Punt is also a receptor for the Activin branch of the TGF-β family [12] so the reduced survival and sterility phenotypes might be associated with disrupting Activin rather than BMP signaling in other tissues.

Potential roles for the lumenal Dpp pool in the wing disc have been noted, particularly in relation to the observed morphogen gradient within the plane of the disc proper epithelium [9]. Recent work using a morphotrap system confirmed the presence of Dpp in the lumen [37], but a variant technique with apicobasal specificity revealed that the Dpp pool within the tissue is required for patterning and growth of the disc [35]. This is consistent with our finding that apicalized Punt is less active, in the sense that Dpp and Punt need to be present in the basolateral cell region for effective signaling. On its face, this behavior is somewhat surprising as the exposure of Punt to the lumenal pool might be predicted to generate ectopic signaling. We note that even clearly apicalized Punt has significant basolateral distribution, so it is not technically possible to assess the function of purely apical Punt. Tkv is present in all membrane domains, so its absence is not likely to preclude apical Dpp signaling. Presently, it is unclear why apical signaling is tepid, but the overall effect of BLT of Punt in the wing disc places it in the proper place to interpret the Dpp morphogen gradient. Further molecular studies are needed to define the *in vivo* spatial and kinetic details of when the Type I and Type II receptors are bound in active signaling complexes, but we note that independent trafficking of Type I and Type II receptors is observed in multiple species [38,39].

Our study of Punt localization and function adds to a collection of studies examining the relevance of restricted signaling in polarized cells. Monolayer cell culture studies have confirmed a tight correlation between confluent epithelia with mature junctions and restricted signaling [40]. In a gastruloid model of embryonic development, interior cells exhibit basolateral restriction of TGF-β receptors whereas cells near the edge of the cell mass lack this polarization and are thus responsive to ligand [41]. An intriguing example of how localization can be harnessed in a developmental context was discovered in the mouse embryo, where basolateral restriction of a BMP receptor in epiblast cells is required to limit SMAD1 activation and thereby support a graded BMP signal output [42].

## A tissue-specific sorting mechanism "reads" the BLT in the wing disc

Polarized membrane traffic is a hallmark of epithelial cells [43]. To situate the basolateral localization of Punt within known traffic networks, we screened *trans*-acting factors to determine which are required for targeting of a functional Punt-GFP protein. With the goal of differentiating between the broad categories of polarized delivery and polarized removal, we perturbed the function of AP-1 and AP-2 Adapter Protein complexes involved with vesicular movement. The AP-1 complex controls delivery of some vesicles to the plasma membrane. Clones with knockdown of AP-1 subunits exhibited apical localization of Punt-GFP, indicating that the

AP-1 complex is required for effective basolateral restriction and suggests that polarized sorting is a key mechanism. In contrast, loss of AP-2–specific subunits or the early endosome regulator Rab5 did not perturb the steady-state basolateral localization of Punt, suggesting that removal of apical Punt is not a significant factor.

The conserved localization behavior directed by the Punt BLT in *Drosophila* wing disc epithelial and mammalian MDCK cells indicates that the machinery to process the BLT instructions are deeply conserved in insect and mammalian genomes. Thus, we were somewhat surprised to find that polarized localization schemes vary between different *Drosophila* epithelia, as seen in the wing disc, salivary gland, and follicle cells. In the context of this cell type–dependent localization, the BLT motif of Punt is therefore best described as an imaginal disc BLT determinant. Sorted proteins often bind directly to AP complex subunits [29], but the tissue-specific nature of the sorting renders this unlikely for the BLT motif. We thus propose that the readout of the *cis*-acting motif depends on the *trans*-acting sorting factors expressed in each epithelial tissue (S9 Fig). Indeed, there are several examples in the broader TGF-β signaling pathway literature consistent with this concept of diversity. Ozdamar and colleagues [44] reported that in NMuMG cells TβRI localizes to junctions and that TβRII dynamically relocalizes from apical puncta toward the junctions upon ligand stimulation. On the BMP side, BMP receptor signaling was reported to be primarily a basolateral process in MDCK cells [45] but to occur both basolaterally and apically in MCF7 cell culture [46]. Extrapolating beyond TGF-β pathway receptors to consider other signaling pathways, we envision a sorting code whereby each epithelial tissue expresses a defined set of sorting factors that read the *cis*-acting determinants of expressed membrane proteins, with the resulting membrane distribution generating a unique signal response profile for each cell type.

# Materials and methods

## Receptor expression constructs and transgenic lines

UAS-Wit-GFP is described in Smith and colleagues [47]. UAS-Punt-GFP DNA was provided by G. Marqués, and random transposition insertions were recovered after injection. Type II chimera constructs were generated by PCR amplification and ligation of the ectoplasmic and cytoplasmic portions of Wit-Flag and Punt-Flag constructs harboring a carboxyl-terminal FLAG epitope [48]. The Wit:Wit construct is a psuedo-WT transgene with a cloning scar that localizes similarly to Wit-GFP. The Wit:Wit, Punt:Wit, and Wit:Punt coding regions were cloned into pUAST and random transposition lines were recovered after injection. Existing UAS fly lines were used to express Tkv-1 [48] and Tkv-2 [49].

Internal deletion of Punt sequences was achieved by PCR amplification of desired cytoplasmic portion (Δ10 or Δ19) and cloning into an HA-Punt vector modified by Quikchange with a *PstI* sequence after the transmembrane boundary. The control HA-Punt-*PstI* protein localizes the same as HA-Punt. Intact coding regions were cloned into pUAST and transgenic lines with random transposition insertions were recovered after injection.

Punt constructs with wholesale substitution of the BLT region were made by dropping oligos into a construct engineered to have restriction sites near the beginning (*PstI*, K185L change) and end (*HindIII*, coding silent) of the BLT region. UAS-attB constructs were injected for recovery of transgenic lines with recombination at the VK20 attP landing site. Similarly, Wit[BLT] variants were made by oligo drop-in to a Wit construct with restriction sites (*NgoMIV* and *HindIII*). UAS-attB constructs were injected to generate transgenic lines with recombination at the VK31and attP1 docking sites.

Smaller Punt Point Mutants (3 amino acids converted to Alanine) were generated by QuickChange site-directed mutagenesis, and longer substitutions were generated by oligo

drop-in as above. UAS-attB constructs were used to generate transgenic lines at the same attP site as UAS-Punt[BLT] above. C-BLT constructs were generated by appending the BLT determinant sequence to Punt-Δ10 or Punt-Δ19 using a 3′ PCR primer, changing the end of coding sequence from..CL* to..CLQAHFNEIPTHEAEITNSSPLL*. Lines with these variants used the VK20 attP site.

Two versions of Punt rescue constructs were used. A larger version was made by recombineering GFP-encoding sequence at the carboxyl terminus of Punt in BAC CH322-93G2. A smaller rescue construct was generated by PCR amplification of 2 genomic regions totaling 7 kb that encompass the *put* gene, with addition of a carboxyl-terminal GFP tag. Sequencing of the entire coding region revealed a serendipitous PCR-generated substitution converting a Glu to Gly within the "THEA" sequence of the BLT motif (EG variant). The WT Punt-GFP construct was made by deploying Quikchange mutagenesis to revert to the expected sequence. The Punt BLT coding region spans 2 exons. The Punt-GFP-Δ19 construct was thus made by successive deletions mediated by QuikChange mutagenesis. Constructs were injected to recover transgenic lines recombined into the attP1 or VK15 landing sites. Rescued animals displayed variable eye defects using the attP1 lines; however, this phenotype tracked with the attP docking site rather than BLT status or basolateral localization (S8 Fig).

### Receptor sequence analysis

Punt homologs were identified by Blast searches against fruit fly Punt (NP_731926.1). Representative receptor sequences from several insect Orders were selected for analysis. The cytoplasmic juxtamembrane region was defined in relation to the predicted transmembrane region and kinase domain, with InterProScan used to mark the boundaries of PHOBIUS "Transmembrane region" and InterPro domain IPR000719. The number of amino acids in the juxtamembrane region varies; spacing was manually adjusted for alignments in Fig 3A. Insect Punt proteins shown are mosquito (Order Diptera, *Anopheles gambiae* sequence XP_311300.4), honeybee (Order Hymenoptera, *Apis mellifera* sequence XP_395928.3), and beetle (Order Coleoptera, *Tribolium castaneum* sequence EEZ97734.1). The juxtamembrane region of Wit is variable in sequence and spacing among insect homologs. The representative honeybee Wit protein in Fig 4 is from *Apis mellifera* (sequence XP_006571217.1).

### MDCK cell culture and protein detection

MDCK cell culture, transfection, protein detection, and imaging were performed as described [8,20]. Briefly, the extracellular pool of receptors was detected by antibody staining in fully polarized MDCK cells after transient transfection. Multiple regions containing positive cells were imaged by confocal microscopy, and xz projections were generated to visualize the apico-basal distribution of the proteins of interest. For Punt detection an HA epitope tag (YDVP-DYALE) was inserted into the extracellular domain after Proline 27. Truncations were generated by PCR amplification of portions of Punt for cloning into pCMV5 or a GM-CSF shuttle vector [20] for expression. Internal deletion constructs were identical to the fly expression versions, but subcloned into pCMV5 for MDCK expression.

### *Drosophila* protein detection

Wing discs and salivary glands from wandering third instar larvae were fixed and subjected to IF staining as described [15]. Overexpressed Receptor-GFP constructs were detected using GFP fluorescence. Other UAS-driven proteins were detected by antibody staining against Punt (rabbit polyclonal, Fabgennix Punt-112-AP), HA (rat monoclonal 3F10, Roche, Pleasanton CA), Flag (mouse monoclonal M2, Sigma, St Louis MO, F3165), Tkv-Pan (rabbit

polyclonal raised and affinity-purified against CVKGFRPPIPSRWQEDDVLAT), or Tkv2 (rabbit polyclonal raised and affinity-purified against SGMEMGSGPGSEGYEDADNEKSK). Endogenous proteins were detected with anti-aPKC (goat polyclonal, Santa Cruz, Santa Cruz CA, sc-216), FasIII (mouse monoclonal, DHSB 7G10), Dlg (mouse monoclonal, DHSB 4F3), or phospho-Mad [50]. Punt-GFP genomic rescue constructs are expressed at low levels, so anti-GFP (Abcam, Cambridge MA, ab6556, preadsorbed against fixed larvae prior to use) was used to image these proteins. Fluorescent secondary antibodies were Alexafluor-488, 568, or 647 (Invitrogen, Waltham MA). DAPI was used as a routine stain for nuclei. For follicle cell imaging, ovaries from yeast-fed females were dissected and fixed as for wing discs. Punt-GFP expression caused severe ovariole dysgenesis with a reduced number of egg chambers.

Most imaging was carried out on a commercial Zeiss LSM710 confocal microscope using a 20X 0.8 N/A objective with an xy resolution of 0.42 microns/voxel, or a 40X 1.20 N/A Water objective with xy resolution of 0.14 microns/voxel (images for Fig 1E–1G, Fig 3B–3D, and Fig 7C and 7D). Excitation was achieved with a 405 nm laser for DAPI, the 488 nm argon laser line for GFP and Alexafluor-488, 561 nm for Alexafluor-568, and 633 nm for Alexafluor-647. Several images were collected using a CARV spinning disc attachment on a Zeiss Axio microscope with a 20x 0.75 N/A objective with an xy resolution of 0.38 microns/voxel (S1E Fig; S3 Fig, S4A–S4C Fig, S5H and S5I Fig). Single sections are shown for *Drosophila* confocal images, except p-Mad images in Fig 6 and in S3 Fig are maximum intensity projections. For compartment-specific p-Mad quantification, the average IF signal in a rectangle of a fixed size was measured for the posterior and anterior compartments using FIJI. Profiles to visualize basolateral receptor distribution relative to markers were generated with the RGBProfilesTool in FIJI, with a line width of 10 pixels or 20 pixels (Fig 3B–3D). Images of wing discs are representative of at least 5 tissues per experimental batch, with each experiment conducted independently at least twice. The "Apical Index" (Fig 5F) from select profiles was calculated as (Punt signal) / (Dlg signal + aPKC signal) for the portion of the profile including the SJ and apical regions, determined based on peak marker staining. Statistical analysis for quantitative data was done in R using Tukey multiple comparison of means test (Tukey HSD).

For western blot analysis, rabbit anti-Punt (as above) and anti-HA (rabbit monoclonal, CST C29F4) were used to probe blots, with anti-rabbit-HRP as a secondary antibody.

### Experimental genotypes

Wing disc expression was achieved with A9-Gal4 (BDSC 8761), *nub*-Gal4 [51], or *vg*-Gal4 (BDSC 8229). *en*-Gal4 (BDSC 30564) was used to drive expression in the posterior compartment, A9-Gal4 was used for salivary gland expression, and GR1-Gal4 (BDSC 36287) was used for follicle cell expression. Endogenous level reporters were used for Punt (this work), Wit [52], and Tkv (Kyoto 115298).

Rescue tests of *wit* by UAS-Wit constructs were conducted by crossing elav-Gal4; *wit^B11*/Balancer females to UAS-Wit[BLT]; *wit^A12*/Balancer males. Progeny were counted by genotype from crosses incubated at the indicated temperature. Rescue tests of *wit* by UAS-Punt constructs used UAS-Punt[BLT], *wit^A12* recombinants. Rescue tests for UAS-Punt used an armGAL4, *put^P1* recombinant crossed to PH[BLT], *put^135* or WF[BLT], *put^135* recombinants, or the UAS elements alone for toxicity tests. Rescue activity of endogenous level constructs were conducted by crossing [Punt transgene]; *put^135*/Balancer flies to *put^P1*/Balancer (1 copy rescue) or [Punt transgene]; *put^P1*/Balancer (2 copy rescue) flies and counting surviving adults. A Rescue Index was calculated based on expected Mendelian frequency of rescued genotype; values greater than 1 indicate better survival than siblings with balancers. For lethal phase analysis, larvae of the "rescue" genotype were identified and transferred to vials, then monitored

for pupariation and adult eclosion rates. Fertility was assessed by test crossing young individual females to control males or individual males to control virgin females, and then scoring for larval progeny. Fisher exact tests for genotype categories were carried out in R.

RNAi hs-flp clones were generated and analyzed as described [23], but with Punt-GFP (BAC construct @VK15) in the background. UAS-RNAi lines used include AP-1/2β (BDSC 28328), AP-1γ (BDSC 27533), AP-2μ (BDSC 28040), AP-2α (BDSC 32866), AP-2σ (BDSC 27322), and Rab5 (BDSC 51847). Activity of these lines was confirmed by production of lethality when crossed to Tub-Gal4/TM6c, Sb, and Tb (Tub-Gal4 from BDSC 5138).

## Supporting information

**S1 Fig. Signaling compartments, delimitation of basolateral and apical membranes in the wing disc, and detection of Tkv at all membrane surfaces. (A)** Simplified model of an epithelium with 2 ligand compartments (green and gray) and 4 distribution patterns of a receptor R based on Ap or BL presentation. Markers for specific membrane regions used in this study [2]. **(B–D)** Confocal imaging of wing discs to reveal distribution of junctions and apicobasal regions. The continuous epithelium of the disc has characteristic folds that bring the apical sides of 2 regions facing toward each other, which appear as "wrinkles" or "folds" in a single confocal plane. **(E–G)** Localization of Tkv relative to membrane domains. Tkv1 isoform overexpressed in the wing disc detected by anti-Tkv(pan) and compared to FasIII SJ marker and aPKC apical marker (E), with profile (E′) recorded at position indicated by yellow arrowheads. Isoforms Tkv1 and Tkv2 from Brummel and colleagues [53] correspond to Tkv-D and Tkv-A isoforms, which differ at the N-terminus. Endogenous Tkv1 and Tkv2 are expressed at low levels in the wing so the IF signal is dominated by the overexpressed protein in the nub-GAL4 domain. Endogenous Tkv detected with anti-Tkv(pan) antibody also showed a general membrane distribution (F and F′). Similar results were obtained for anti-GFP staining of a *tkv* allele expressing Tkv-YFP (G and G′). Ap, apical; BL, basolateral; IF, immunofluorescence; SJ, septate junction.
(TIFF)

**S2 Fig. Punt cytoplasmic deletion series in MDCK cells reveals conserved localization behavior of insect and mammalian receptors, but relying on different sequences.** Schematic of the component segments of receptors tested for basolateral restriction is shown to the left, and a representative xz confocal projection to the right. In the diagram, the ectodomain is to the left of the membrane, which is marked by vertical red bar, and the cytoplasmic domain is to the right. In the images, apical is up and nuclei are stained in blue with DAPI. The β protein from Murphy and colleagues displays basolateral localization of a chimeric receptor containing the TβRII cytoplasmic domain. The BA shuttle lacks the required LTA motif, depicted by the red rectangle within the green TβRII protein, and thus displays an unrestricted apicobasal distribution (red arrow points to apical staining). WT Punt is also restricted to the basolateral membrane domains, as are chimeric proteins containing various cytoplasmic portions of Punt. The corresponding LTA position, shown by the red rectangle in the blue Punt protein, is not required for the basolateral localization since progressive carboxyl-terminal truncations lacking this region retain the activity. The shortest truncation tested possessed only a short stretch of the cytoplasmic portion of Punt. There are di-leucine elements at positions 196+197 and 204+205, but neither is required for basolateral restriction since the shortest truncation ending at amino acid 195 retains basolateral localization. MDCK, Madin-Darby canine kidney; TβRII, TGF-β Type II receptor; WT, wild-type.
(TIFF)

**S3 Fig. Overexpressed Punt leads to ectopic P-Mad with or without BLT motif. (A)** *vg*-GAL4 expression pattern in third instar wing disc as shown by UAS-nlsGFP reporter (green; DAPI in blue), single confocal section. **(B)** p-Mad detection pattern in WT disc, maximum intensity projection. **(C–E)** p-Mad signal upon overexpression of intact Punt (C) and protein variants missing part (D) or all (E) of the BLT motif. p-Mad in white, DAPI in blue. Punt (Punt-*PstI*) and both deletions generate ectopic pMad in the pouch and at anterior side of disc. Widened discs are consistent with overgrowth caused by excess Dpp/BMP signaling (compare widths in C–E to control discs in A and B). Note that these constructs are inserted at random genomic positions with different expression levels, so the degree of ectopic signaling cannot be correlated with protein deletions in this context. Anterior is to the left in all images. Scale bars: 50 μm. BLT, basolateral targeting; WT, wild-type.
(TIFF)

**S4 Fig. Deletion of the Punt BLT region leads to loss of basolateral restriction in the wing disc. (A–C)** Punt-HA variants expressed with *vg*–GAL4 detected with an antibody for the extracellular HA epitope. White arrow in A indicates extent of Punt detection signal, which remains basal to the junction. Yellow arrows in B and C indicate ectopic staining apical to the junctions. **(D–F)** Punt-HA variants expressed with *nub*-GAL4 detected with cytoplasmic anti-Punt antibody. Punt detection is absent at or apical to the SJs (D). PuntΔ10 shows significant overlap with the SJ but in many areas is excluded from the apical membrane (E). PuntΔ19 shows significant overlap with junction and apical regions (F). Single prime images are close-up views; yellow arrowheads indicate position and direction of profiles displayed in double prime panels. Scale bars: 50 μm A–C, 100 μm D–F, 15 μm D′–F′. BLT, basolateral targeting; SJ, septate junction.
(TIFF)

**S5 Fig. Positional and mutational analysis of the Punt BLT. (A–C)** RGB profiles for Punt BLT region substitution constructs, with corresponding Fig 4 panels indicated. The control Punt[Punt] and Punt[Apis] proteins are largely confined to the lateral membranes, and excluded from junction and apical membrane domains (A, B). Punt[Wit], a Punt protein harboring a stretch of Wit amino acids only in the juxtamembrane region, showed unrestricted distribution with an apical preference (C). **(D–F)** Localization of point mutant versions of Punt. The EEE, FNE, and EIP and triple point mutants generally retained basolateral presentation, but the EIP mutant had variable regions of apicalization (compare profile examples a and b below F). Direction and position of profile line are indicated by yellow arrowheads, as indicated in D inset. **(G)** Sequence of BLT point mutants; all include the K and L mutation from *PstI* cloning scar. **(H, I)** Addition of the BLT to carboxyl terminus of otherwise apicalized Punt proteins. L3 wing discs dissected and stained for Punt and apical aPKC. Constructs had either 10 or 19 residues from the BLT region deleted and the BLT appended to the carboxyl terminus. When the insect-conserved residues of the BLT region were deleted (Δ10) and the BLT was added carboxyl-terminally, apical mislocalization of Punt was still observed (H). Deletion of the entire BLT region (Δ19) and addition of the BLT carboxyl-terminally resulted in apical mislocalization of Punt, as seen by the yellow stripe of colocalization (I). H′ and I′ are enlarged to better show the apical stripe. Scale bars: 100 μm D–F, H, I; 50 μm H′, I′. BLT, basolateral targeting.
(TIFF)

**S6 Fig. Apicobasal distribution of Wit protein variants. (A–D)** Profiles of Wit proteins shown in Fig 4. Anti-FLAG IF in the wing disc displayed a significant background signal, with enrichment at the apical membrane (A). Wit[Wit] was detected in all regions, with an

enrichment overlapping the apical membrane (B). Wit[Punt], which is a Wit protein with Punt amino acids only in the juxtamembrane region, had a localization more like Punt than Wit (C). Deletion of the Wit juxtamembrane region did not alter the protein's apicobasal distribution (D).
(TIFF)

**S7 Fig. Rescue attempts and toxicity of UAS-Punt[BLT] and UAS-Wit[BLT] constructs.**
**(A)** *wit* mutants were rescued to viability by overexpression of Punt or Wit with *elav*-GAL4, regardless of which version of the juxtamembrane sequence was used (see key). **(B)** *arm*-GAL4 driving UAS-Punt proteins did not restore viability of *put*$^{135/P1}$ mutants. PH[xx] indicates Punt-HA harboring the BLT sequence indicated in brackets. WF[xx] indicates Wit-Flag harboring the BLT sequence indicated in brackets. Toxicity tests in a *put* heterozygote background show significant lethality from ectopic expression of Punt[Punt] and Punt[Apis], but not Punt[Wit]. **(C)** UAS-Wit proteins also did not rescue *put* mutants, regardless of the BLT status. Toxicity tests showed that in this case arm>Wit did not lead to lethality. N/D: Not determined. **(D)** Punt[Wit] was less toxic than Punt[Punt], as shown by differential temperature sensitivity of male (more expression from A9-Gal4) and female (less expression) animals. The data underlying the graph shown in this figure can be found in S1 Data. BLT, basolateral targeting.
(TIFF)

**S8 Fig. Expression pattern and eye defects arising from specific combination of attP site and rescue construct.** Patterned Punt-GFP detection was observed with the 7 kb rescue construct recombined into the attP1 docking site. However, a larger rescue construct at the same site showed uniform staining in the wing disc, as did the 7 kb construct at the VK15 docking site. There is thus a genome region influence only on the smaller construct. Variable eye defects (some adult eye tissue missing) were also only seen with the 7 kb construct at attP1, likely due to expression influences.
(TIFF)

**S9 Fig. Interrogation of cellular trafficking machinery suggests a sorting mechanism operates on the BLT. (A)** Two general alternative mechanisms to achieve steady state basolateral restriction are presented, the Sorting Model and the Turnover Model. To differentiate between them, proteins required for sorting and delivery versus endocytosis were knocked down by RNAi to determine if localization of Punt-GFP was altered. Summarized results are shown in the table **(B)**. **(C)** Based on results in *Drosophila* epithelia showing that the Punt BLT is a tissue-specific basolateral determinant, we propose a sorting paradigm utilizing tissue-specific loaders to direct receptors with targeting motifs to general or restricted membrane domains. BLT, basolateral targeting; RNAi, RNA interference.
(TIFF)

**S1 Data. Numerical data.**
(XLSX)

## Acknowledgments

We thank Guillermo Marqués for the Punt-GFP construct, providing Wit-GFP flies prior to publication, and sharing project and manuscript advice; Mihaela Serpe for initial characterization of Punt-GFP flies; Melissa Ritter for cloning and analysis of Punt and Wit function; and Autumn Pace for preliminary studies of AP-1 components in the wing. We thank Myung-Jun Kim, Amanda Neisch, and Heidi Bretscher for helpful manuscript feedback.

## Author Contributions

**Conceptualization:** Aidan J. Peterson, Edward B. Leof, Michael B. O'Connor.

**Data curation:** Aidan J. Peterson, Stephen J. Murphy, Melinda G. Mundt, MaryJane Shimell.

**Formal analysis:** Aidan J. Peterson, Stephen J. Murphy.

**Funding acquisition:** Michael B. O'Connor.

**Investigation:** Aidan J. Peterson, Stephen J. Murphy, Melinda G. Mundt, MaryJane Shimell, Edward B. Leof.

**Methodology:** Aidan J. Peterson, Melinda G. Mundt, Edward B. Leof.

**Project administration:** Michael B. O'Connor.

**Validation:** Stephen J. Murphy, MaryJane Shimell, Edward B. Leof.

**Writing – original draft:** Aidan J. Peterson.

**Writing – review & editing:** Stephen J. Murphy, Melinda G. Mundt, MaryJane Shimell, Edward B. Leof, Michael B. O'Connor.

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
