## [Editor Report · Decision Letter 0]

6 Oct 2021

Dear Dr O'Connor, 

Thank you for submitting your manuscript entitled "A juxtamembrane basolateral targeting motif regulates TGF-β receptor signaling in Drosophila" for consideration as a Research Article by PLOS Biology.

Your manuscript has now been evaluated by the PLOS Biology editorial staff as well as by an academic editor with relevant expertise and I am writing to let you know that we would like to send your submission out for external peer review. Please note, however, that the outcome of our discussion of your manuscript is that we have some reservations as to the overall advance offered by your data. We would need to be persuaded by the reviewers that the paper has the potential after revision to offer the significant strength of advance that we require for publication in order to pursue it further for PLOS Biology.

Before we can send your manuscript to reviewers, we need you to complete your submission by providing the metadata that is required for full assessment. To this end, please login to Editorial Manager where you will find the paper in the 'Submissions Needing Revisions' folder on your homepage. Please click 'Revise Submission' from the Action Links and complete all additional questions in the submission questionnaire.

Once your full submission is complete, your paper will undergo a series of checks in preparation for peer review. Once your manuscript has passed the checks it will be sent out for review. 

If your manuscript has been previously reviewed at another journal, PLOS Biology is willing to work with those reviews in order to avoid re-starting the process. Submission of the previous reviews is entirely optional and our ability to use them effectively will depend on the willingness of the previous journal to confirm the content of the reports and share the reviewer identities. Please note that we reserve the right to invite additional reviewers if we consider that additional/independent reviewers are needed, although we aim to avoid this as far as possible. In our experience, working with previous reviews does save time. 

If you would like to send your previous reviewer reports to us, please specify this in the cover letter, mentioning the name of the previous journal and the manuscript ID the study was given, and include a point-by-point response to reviewers that details how you have or plan to address the reviewers' concerns. Please contact me at the email that can be found below my signature if you have questions. 

Please re-submit your manuscript within two working days, i.e. by Oct 08 2021 11:59PM.

Kind regards,

Ines

--

Ines Alvarez-Garcia, PhD

Senior Editor

PLOS Biology

---

## [Decision Letter · Decision Letter 1]

29 Nov 2021

Dear Dr O'Connor,

Thank you for submitting your manuscript entitled "A juxtamembrane basolateral targeting motif regulates TGF-β receptor signaling in Drosophila" for consideration as a Research Article at PLOS Biology. Thank you also for your patience as we completed our editorial process, and please accept my apologies for the delay in providing you with our decision. Your manuscript has been evaluated by the PLOS Biology editors, an Academic Editor with relevant expertise, and by three independent reviewers.

As you will see, the reviewers find your conclusions novel and interesting, but they also raise some concerns that need to be addressed before we can consider the manuscript for publication. Both Reviewers 1 and 3 think that the evidence demonstrating Punt basolateral localisation is not convincing and that you need to include cross-sections and perform other experiments to strengthen this result. Reviewer 1 also thinks that you should consider and discuss other potential interpretations of the results. Reviewer 2 has several good points regarding the nomenclature. After discussing the reviews with the Academic Editor, we would like to consider a revised version of the manuscript that addresses all these points, especially the demonstration of the basolateral localisation with high resolution confocal Z-sections or histological tissue sections. Regarding the potential alternative interpretations, we would like you to describe better your point of view and consider and discuss the alternatives.

In light of the reviews (attached below), we will not be able to accept the current version of the manuscript, but we would welcome re-submission of a revised version that takes into account the reviewers' comments. We cannot make any decision about publication until we have seen the revised manuscript and your response to the reviewers' comments. Your revised manuscript is also likely to be sent for further evaluation by the reviewers.

We expect to receive your revised manuscript within 3 months. 

**IMPORTANT - SUBMITTING YOUR REVISION**

3. Resubmission Checklist

a) *PLOS Data Policy*

b) *Published Peer Review*

d) *Blurb*

Please also provide a blurb which (if accepted) will be included in our weekly and monthly Electronic Table of Contents, sent out to readers of PLOS Biology, and may be used to promote your article in social media. The blurb should be about 30-40 words long and is subject to editorial changes. It should, without exaggeration, entice people to read your manuscript. It should not be redundant with the title and should not contain acronyms or abbreviations. For examples, view our author guidelines: https://journals.plos.org/plosbiology/s/revising-your-manuscript#loc-blurb

Sincerely,

Ines

--

Ines Alvarez-Garcia, PhD

Senior Editor

PLOS Biology

Reviewers' comments

Rev. 1:

In this manuscript, O'Connor and colleagues utilize the wing epithelium of Drosophila to analyze the membrane localization of Dpp receptors Tkv, Punt and Wit, to identify the evolutionary conserved relevant regions aimed at targeting these receptors to the apical or basolateral sides of the epithelium and to characterize the functional relevance of this targeting to Dpp signaling and patterning. The paper is well written, the main messages are subdivided into independent sections and figures, and the biological question is timely and relevant in the context of signaling. Authors focus on the relevance of the localization of Punt in particular, and identify a 19 aa-long region required for basolateral localization of the protein in the wing epithelium. I have two major concerns (one technical and one more profound):

(1) Basolateral localization of Punt is not convincing.

- Figure 1C: The basolateral localization of Punt-GFP is not convincing. Thus, "Overexpressed Punt protein exhibited basolateral restriction, with nearly all of the protein staying basal to the septate junction" is not supported by the data. Figure 1C: no labeling of FasIII was shown in this panel. I would suggest authors to perform cross-sections of the epithelium to support better evidence for the preferential localization of Punt (basolateral) and Wit (apical).

- Figure 2F-G: authors should label apical and basal sides of the epithelial sheet.

- Figure 3B-D: Again, authors should perform cross-sections of the epithelium to support better evidence for the preferential localization of wt Punt (basolateral) when compared to truncated versions of the protein. Data in Figure 3-S2 is much clearer where the profile of panel D'' is perhaps the most convincing to state the basolateral localization of wt Punt. The apical enrichement of Delta-10 in panel E'' is clear on one side but on the other. Authors should change it with more convincing data. The profile in panel F'' is very convincing. Why is the Delta-BLT and not the full cytoplasmic truncation localized apically in MDCK cells?

- Figure 4: The profile of Punt-Apis should be shown to demonstrate its BL localization. Authors use the Punt-Wit quimeric protein and its apical enrichment to state that "basolateral targeting is a specific function of Punt juxtamembrane sequences". These data indicate that the cytoplasmic domain of Wit targets the protein to the apical side of the epithelium. Authors should make the Punt(Apis) lacking the cytoplasmic domain (or the BLT region) to experimentally demonstrate their statement. Data in panel 4B are shown fly Punt? If so, this should be stated as Punt (Drosophila). Profiles of panels I, J should be included.

- Figures 5E-J: Profiles of panels 5E-G required. It's sad to see authors do not use these constructs (instead of overexpression experiments) to present evidence of the requirement of these 19 aa to localize Punt basolaterally.

(2) Functional relevance of basolateral localization of Punt. Perhaps the most important data in this regard are shown in Figure 6F-M where a rescue construct of Punt lacking the BLT region (the one required to target Punt basolaterally) is able to almost completely rescue Dpp signaling and the adult wing phenotype. In this context, I wonder whether these data indicate that Punt localization is not relevant (which might frustrating and unexpected) or simply because the BLT region is not the one that targets Punt basolaterally but the one that excludes it from being localized apically. In order terms, all data can be explained by "ectopic" expression of Punt in the absence of BLT and not by loss of Punt basolaterally (where it binds the Dpp ligand. Thus, perhaps the major conclusion of these data is that ectopic localization of Punt in the apical domain is not relevant. I agree, though, that the BLT is required for animal fitness (as shown in Figure 5), but it's not demonstrated whether reduced fitness is a consequence of the protein being misplaced in the epithelium.

Minor issues: data presented in Figure 5A-C are out of context (from my point of view) and some of these data (eg- lethality) not quite conclusive. Labels of apical vs basal should be included in all the figures. Data presented in 7E-H not very convincing. Some of the last figures are difficult to interpret.

Rev. 2:

Using Drosophila as model system, the authors show the differential localization, i.e. basolateral versus apical, of two Dpp receptors, Punt and Wit. This observation provides the basis for their subsequent experiments that lead to the definition of the basolateral targeting (BLT) sequence in Punt, and of the role of this sequence, and of basolateral signaling, in Dpp's signaling activity in fly development.

Overall: I did not evaluate this manuscript from the perspective of a Drosophila developmental geneticist (not my area), but rather did so from my broad knowledge of TGF-b family signaling and biology.

This is a solid and straightforward story that is logically pursued, scientifically sound, clearly presented, and with the conclusions supported by the data. My comments relate to the presentation.

1. In the title and throughout the manuscript the authors talk about TGF-b receptors and their signaling. This is not appropriate, since Dpp is a homolog of BMP2 and BMP4. I realize that Drosphila labs and journals (editors) often use the term TGF-b when referring to Dpp, and that this comment may come over as annoying. However, considering Dpp as TGF-b is not only incorrect, but it also creates substantial confusion. For example, I have been in a thesis committee for a graduate student in a hybrid Drosphila/mouse lab, and the graduate student was pursuing TGF-b studies in mouse cell differentiation based on Drosophila observations related to Dpp, which was "advertised" as the homolog of TGF-b. This led to at least a year of wasted effort and a need to redesigning the thesis proposal with BMP (not TGF-b) being the homolog of Dpp. And of course BMPs often have effects that drastically differ from TGF-b.

2. Please rename "superfamily" into "family". The TGF-b family is a family, not a superfamily. Again this comment that may be dismissed as annoying.

3. The Discussion often reads as a rehashing of the Results. No need to repeat what you already wrote in Results. Make it shorter and eliminate the retelling of the Results, while keeping the Discussion to a critical and insightful discussion of the Results. As it currently stands, the Discussion does not give me much new insight that I did not find in Results.

Rev. 3:

TGF-b signaling is tightly controlled at multiple levels. Spatial coordination of TGF-b ligands and receptors is one of the critical regulations to elicit a proper signaling output. In this study, Peterson et al. observed a restricted basolateral membrane localization of the TGF-b type II receptor Punt in Drosophila wing disc epithelia. They reveal a novel basolateral targeting sequence (BLT) at the juxtamembrane position of Punt by comparing amino acid sequences between TGF-b receptors and by analyzing a series of deletion mutants. punt mutants lacking the BLT showed wing developmental defects, lethality, and female sterility. Based on these results, they concluded that proper basolateral localization of Punt is essential for optimal TGF-b signal transduction in Drosophila.

Major comments:

1. The results supporting the requirement of Punt's BLT in wing development are convincing. However, whether and how Punt mislocalization leads to sterility and lethality is unclear. Indeed, only wing disc epithelia show the restricted Punt localization. Thus, the BLT deletion may differentially influence Punt activity in other tissues. The authors should directly address this discrepancy.

2. The wing folding is useful for visualizing apicobasal protein localization. However, readers with untrained eyes may have difficulty interpreting the images from the folding regions. Therefore, please also provide conventional crosssection images. In addition, it will be critical to examine endogenous receptor localizations since antibodies and transgenic lines to visualize their endogenous expression are available.

Minor comments:

1. The authors stated that the Punt deletions (delta 10 and 19) did not destroy its signal transduction activity. However, as shown in Figure 3-figure supplement 1, the delta 19 mutant seems to have a dominant-negative effect.

2. The authors used several transgenic lines and distinct detection methods to visualize Punt localizations but did not clearly describe how Punt proteins were detected in some figures.

3. No data supports the conserved mutant protein's varied apicobasal localizations (page 13, line 21-23; Figure 4F).

4. Please provide sample sizes for all quantifiable data.

5. Do not use abbreviations without explanation.

6. A large BAC... (data not shown). > A large BAC... (Figure 5-supplement 2). (page 19, line 14)

---

## [Decision Letter · Decision Letter 2]

14 Apr 2022

Dear Dr O'Connor,

Thank you for submitting your revised Research Article entitled "A juxtamembrane basolateral targeting motif regulates signaling through a TGF-β pathway receptor in Drosophila" for publication in PLOS Biology. I have now obtained advice from two of the original reviewers and have discussed their comments with the Academic Editor. 

Based on the reviews (attached below), we will probably accept this manuscript for publication, provided you satisfactorily address the remaining points raised by the reviewers. Please also make sure to address the following data and other policy-related requests stated below.

We expect to receive your revised manuscript within two weeks. 

*Published Peer Review History*

*Press*

Sincerely,

Ines

Ines Alvarez-Garcia, PhD,

Senior Editor,

ialvarez-garcia@plos.org,

PLOS Biology

DATA POLICY:

Thank you for providing the data underlying all the graphs shown in the figures. However, I have two pending requests:

- Please provide the data underlying the graph shown in Fig. 5I.

- Please also ensure that figure legends in your manuscript include information on where the underlying data can be found.

DATA NOT SHOWN?

Reviewers' comments

Rev. 1:

Authors have addressed my concerns satisfactorily.

However, important issues that should be seriously addressed by the authors before considering it for publication include:

(1) Statistics and description of intensity profiles and pMAD signal intensity quantification are unfortunately lacking and should be included. Software used, details on statistics and a table summarizing all these data should be included.

(2) The lines where the profiles come from should be depicted in all figures.

(3) Position and size of labels and even font type and size appear to be rather chaotic.

(4) Scale bars for all panels should be provided

(5) Histograms and plots should be "aesthetically" improved.

Rev. 2:

This is a nice and solid paper, worth publishing and reading.

Minor: page 20, line 22: "add" should read "adds".

---

## [Editor Report · Decision Letter 3]

4 May 2022

Dear Dr O'Connor,

On behalf of my colleagues and the Academic Editor, Konrad Basler, I am pleased to say that we can in principle accept your Research Article entitled "A juxtamembrane basolateral targeting motif regulates signaling through a TGF-β pathway receptor in Drosophila" for publication in PLOS Biology, provided you address any remaining formatting and reporting issues. These will be detailed in an email that will follow this letter and that you will usually receive within 2-3 business days, during which time no action is required from you. Please note that we will not be able to formally accept your manuscript and schedule it for publication until you have completed any requested changes.

PRESS

Sincerely, 

Ines

--

Ines Alvarez-Garcia, PhD 

Senior Editor 

PLOS Biology
